# Combined and differential roles of ADD domains of DNMT3A and DNMT3L on DNA methylation landscapes in mouse germ cells

Naoki Kubo [1,4] ✉, Ryuji Uehara[1], Shuhei Uemura [1,2], Hiroaki Ohishi [3], Kenjiro Shirane [2] & Hiroyuki Sasaki [1] ✉

DNA methyltransferase 3A (DNMT3A) and its catalytically inactive cofactor DNA methyltransferase 3-Like (DNMT3L) proteins form functional hetero-tetramers to deposit DNA methylation in mammalian germ cells. While both proteins have an ATRX-DNMT3-DNMT3L (ADD) domain that recognizes histone H3 tail unmethylated at lysine-4 (H3K4me0), the combined and differential roles of the domains in the two proteins have not been fully defined in vivo. Here we investigate DNA methylation landscapes in female and male germ cells derived from mice with loss-of-function amino acid substitutions in the ADD domains of DNMT3A and/or DNMT3L. Mutations in either the DNMT3A-ADD or the DNMT3L-ADD domain moderately decrease global CG methylation levels, but to different degrees, in both germ cells. Furthermore, when the ADD domains of both DNMT3A and DNMT3L lose their functions, the CG methylation levels are much more reduced, especially in oocytes, comparable to the impact of the *Dnmt3a/3L* knockout. In contrast, aberrant accumulation of non-CG methylation occurs at thousands of genomic regions in the double mutant oocytes and spermatozoa. These results highlight the critical role of the ADD-H3K4me0 binding in proper CG and non-CG methylation in germ cells and the various impacts of the ADD domains of the two proteins.

DNA methylation in the mammalian germ line is fundamental for spermatogenesis, paternal and maternal genomic imprinting, and embryonic development after fertilization[1–5]. DNA methyltransferase 3A (DNMT3A) and its catalytically inactive cofactor DNA methyltransferase 3-Like (DNMT3L) proteins form a linear heterotetramer to establish germ line DNA methylation landscapes[6,7]. The regulation of the proper distribution of DNA methylation is heavily dependent on the histone modification states recognized by the DNMT3A-DNMT3L complex in germ cells[1,8]. While the Pro-Trp-Trp-Pro (PWWP) domain and ubiquitin-dependent recruitment (UDR) domain of DNMT3A

respectively interact with di-/tri-methylated histone H3 lysine-36 (H3K36me2/3) and monoubiquitylated histone H2A lysine-119 (H2AK119ub) to promote DNA methylation[9–15], the ATRX-DNMT3-DNMT3L (ADD) domains carried by both DNMT3A and DNMT3L interact with unmethylated histone H3K4 (H3K4me0) to prevent de novo DNA methylation at H3K4me2/3 marked regions[16–19]. A recent structural analysis showed that the ADD domain of DNMT3A releases autoinhibition of its own methyltransferase activity upon H3K4me0 binding but shows reduced interactive activity with H3K4me3 marked chromatin[20]. In addition, mutations in the DNMT3A-ADD domain have

[1]Division of Epigenomics and Development, Medical Institute of Bioregulation, Kyushu University, Fukuoka 812-8582, Japan. [2]Department of Genome Biology, Graduate School of Medicine, Osaka University, Osaka 565-0871, Japan. [3]Division of Gene Expression Dynamics, Medical Institute of Bioregulation, Kyushu University, Fukuoka 812-8582, Japan. [4]Present address: Department of Experimental Genome Research, Research Institute for Microbial Diseases, Osaka University, Suita, Osaka 565-0871, Japan. ✉e-mail: naoki.kubo@bioreg.kyushu-u.ac.jp; hsasaki@bioreg.kyushu-u.ac.jp

been found in various human diseases[21–24], suggesting a certain impact of dysfunction of the ADD domain in mammalian cells. Consistent with these findings, a previous study reported a partial reduction of global DNA methylation level in mouse neonatal prospermatogonia upon a loss-of-function mutation in the DNMT3L-ADD domain[25], and we recently reported that a similar loss-of-function mutation in the DNMT3A-ADD domain also causes a partial reduction in global DNA methylation in mouse oocytes[26]. These results suggest that a loss of ADD-H3K4me0 interaction in either DNMT3A or DNMT3L affects the de novo DNA methylation process in the mammalian germ lines. However, the combined and differential roles of the ADD domains of the two proteins in the same cell and in the different sexes have not been clarified.

To gain a better understanding of the role of the DNMT3A-ADD and DNMT3L-ADD domains in vivo, we investigate female and male germ cells derived from mice with loss-of-function amino acid substitutions in the respective ADD domains. We observe distinct impacts of the DNMT3A-ADD and DNMT3L-ADD mutations on the CG methylation landscapes and drastic alterations of CG and non-CG methylation levels in the double mutant mice. Our findings uncover combined and differential roles of the ADD domains of the DNMT3A-DNMT3L complex in establishment of the proper CG and non-CG methylation landscapes in the mouse germ lines.

## Results

### Mice carrying loss-of-function ADD mutations in DNMT3A and DNMT3L

We recently generated mice carrying aspartic-acid-to-alanine substitutions at both D525 and D527 in the DNMT3A-ADD domain[26]. In this study, we have additionally generated mice carrying an aspartic-acid-to-alanine substitution at D124 in the DNMT3L-ADD domain using CRISPR/Cas9 (see Materials and Methods) (Fig. 1a, Supplementary Fig. 1a, Supplementary Data 1). The amino acid substitutions were previously shown to reduce binding of the ADD domains to H3K4me0 in in vitro binding assays[20,25,26]. The mutated alleles are respectively referred to as *Dnmt3a^ADD* and *Dnmt3L^ADD*. We then generated double homozygous [*Dnmt3a^ADD/ADD*, *Dnmt3L^ADD/ADD*] mutant mice by crossing double heterozygous [*Dnmt3a^ADD/+*, *Dnmt3L^ADD/+*] female and male mice. We confirmed the expression of the mutated DNMT3A and DNMT3L proteins in [*Dnmt3a^ADD/ADD*, *Dnmt3L^ADD/ADD*] mice by western blotting, as was observed in the respective homozygous mice[25,26] (Fig. 1b). Double homozygous [*Dnmt3a^ADD/ADD*, *Dnmt3L^ADD/ADD*] mice were obtained at a rate of 4.5% (female 2.2%, male 2.3%), which is not much different from the expected Mendelian ratio (6.25%) and the observed ratio of the wild-type mice (6.0%) (Fig. 1c). [*Dnmt3a^ADD/ADD*, *Dnmt3L^ADD/ADD*] mice were viable, but they showed a slightly decreased body weight (Fig. 1d, Supplementary Fig. 1b).

We then examined the fertility of the mutant mice by crossing them with wild-type C57BL/6J mice. We previously reported that *Dnmt3a^ADD/ADD* females were not able to give viable pups due to developmental arrest and potential delivery problems[26] (Fig. 1e). In contrast, *Dnmt3L^ADD/ADD* females successfully delivered viable pups with a litter size comparable to that from wild-type females (Fig. 1e). Regarding male mutant mice, *Dnmt3a^ADD/ADD* males showed normal fertility, in contrast to *Dnmt3a^ADD/ADD* females, and *Dnmt3L^ADD/ADD* males produced a slightly smaller number of viable pups than wild-type males, as reported previously[25]. In the case of [*Dnmt3a^ADD/ADD*, *Dnmt3L^ADD/ADD*] mice, neither males nor females gave viable pups.

Given the infertility of the double homozygous mice, we next investigated their gonads. The overall ovary size was similar between the double homozygous and wild-type females (Fig. 1f), and we were able to obtain over 20 fully grown oocytes (FGOs) per [*Dnmt3a^ADD/ADD*, *Dnmt3L^ADD/ADD*] ovary. There was no discernible change in size or gross morphology of these FGOs compared with wild-type ones (Supplementary Fig. 1c). In contrast, a significant size reduction was observed in the testes of the [*Dnmt3a^ADD/ADD*, *Dnmt3L^ADD/ADD*] males (Fig. 1f), as

reported previously for the *Dnmt3L^ADD/ADD* males[25]. The number of spermatids was reduced in the double homozygous seminiferous tubules, and spermatozoa obtained from the epididymis showed severe impairment in motility and viability compared with wild-type, *Dnmt3a^ADD/ADD*, or *Dnmt3L^ADD/ADD* spermatozoa (Supplementary Fig. 1d, e, Supplementary Movies 1–4).

### Altered accumulation of DNMT3A^ADD and DNMT3L^ADD proteins in the FGO nucleus

Next, we explored the intranuclear localization of DNMT3A and DNMT3L proteins in FGOs upon the loss of function of their ADD domains. Immunofluorescence staining of wild-type FGOs showed that DNMT3A and DNMT3L signals were present in the nucleus with greater intensities in 4,6-diamidino-2-phenylidole (DAPI)-dense heterochromatin[27–29] (Fig. 2a, b, Supplementary Fig. 2a, b), consistent with their role in de novo methylation during oogenesis. However, their relative fluorescence intensities in the heterochromatin were significantly reduced in *Dnmt3a^ADD/ADD* and *Dnmt3L^ADD/ADD* FGOs (Fig. 2a, b, Supplementary Fig. 2a, b). Notably, in *Dnmt3a^ADD/ADD* FGOs that still have wild-type DNMT3L, the fluorescent signals of both proteins were reduced (Supplementary Fig. 2a, b). Conversely, a significant reduction of the DNMT3A signals was observed in the heterochromatin of *Dnmt3L^ADD/ADD* FGOs. The degrees of signal reductions of both proteins in these single homozygous mutants were comparable with that observed in [*Dnmt3a^ADD/ADD*, *Dnmt3L^ADD/ADD*] FGOs (Fig. 2b), suggesting that there is little, if any, additive effect of the ADD mutations on the altered localization of the DNMT3A-DNMT3L complex. Based on the decreased DNMT3A signal throughout the nucleus (Supplementary Fig. 2c), it is possible that the loss of ADD-H3K4me0 interaction affects not only intranuclear localization but also nuclear retention.

### Impact of the loss-of-function ADD mutations in DNMT3A and/or DNMT3L on the CG methylation landscape in FGO

We next investigated the impact of *Dnmt3a^ADD/ADD* and/or *Dnmt3L^ADD/ADD* mutations on the DNA methylation landscape in oocytes by whole-genome bisulfite sequencing (WGBS) (Supplementary Data 2). Although the accumulation of the DNMT3A-DNMT3L complex in the nucleus appeared to be similarly altered in all single and double homozygous FGOs, as described above (Fig. 2), the degree of global CG methylation reduction varied among the mutants. In general, the impact of the DNMT3L-ADD mutation was less severe than that of the DNMT3A-ADD mutation in FGOs (*Dnmt3L^ADD/ADD*, 26.6%; *Dnmt3a^ADD/ADD*, 17.6%) (Fig. 3a, b), and the differences were largely uniform across the genome (Fig. 3c). Notably, when both DNMT3A-ADD and DNMT3L-ADD domains had mutations, the global CG methylation level was much more reduced ([*Dnmt3a^ADD/ADD*, *Dnmt3L^ADD/ADD*], 6.2%), and to our surprise, comparable with that of DNMT3A or DNMT3L knockout FGOs[30] (Fig. 3a, b, Supplementary Fig. 3a). However, there were regions that showed not much reduced or even higher CG methylation levels in the double mutant FGOs (Fig. 3a, see below for more details). Next, we analyzed the CG methylation levels in maternal and paternal imprinting control regions (ICRs) that are normally fully CG-methylated in wild-type FGOs and spermatozoa, respectively. In FGOs, the degree of CG methylation reduction was greater in the order of [*Dnmt3a^ADD/ADD*, *Dnmt3L^ADD/ADD*] (81-92% reduction), *Dnmt3a^ADD/ADD* (35-66% reduction), and *Dnmt3L^ADD/ADD* (14-39% reduction) at all analyzed maternal ICRs (Fig. 3d). Furthermore, CG methylation at repetitive elements such as LINE, LTR, and SINE as well as repetitive element subfamilies (classification defined in RepeatMasker[31]) was similarly affected in these mutant FGOs (Fig. 3e, Supplementary Fig. 3b).

### Impact of the loss-of-function ADD mutations in DNMT3A and/or DNMT3L on the CG methylation landscape in spermatozoa

We next analyzed the CG methylation landscapes of *Dnmt3a^ADD/ADD* and/or *Dnmt3L^ADD/ADD* spermatozoa. The *Dnmt3a^ADD/ADD* spermatozoa

showed a more severe reduction in global CG methylation levels than the $Dnmt3L^{ADD/ADD}$ spermatozoa ($Dnmt3L^{ADD/ADD}$, 75.4%; $Dnmt3a^{ADD/ADD}$, 66.3%), which was the same trend observed in the corresponding mutant FGOs (Fig. 4a, b). However, certain genomic regions, such as the lamina-associated domains, which mostly overlap with the gene desert regions, exhibited more severe CG methylation reduction in $Dnmt3L^{ADD/ADD}$ spermatozoa (Fig. 4c, Supplementary Fig. 4a). In addition, the paternal ICR of $Rasgrf1$ was more severely affected in $Dnmt3L^{ADD/ADD}$ spermatozoa (Fig. 4d), consistent with the fact that the

methylation of the $Rasgrf1$ ICR is heavily dependent on DNMT3C, which also forms a complex with DNMT3L[32]. When both DNMT3L-ADD and DNMT3A-ADD domains had the mutations, the CG methylation level was more reduced, but only to a 50.4% level (Fig. 4a, b), in contrast to the more severe reduction to a 6.2% level observed in double mutant FGOs (Fig. 3a, b). This suggests that the ADD domains would have less important roles in CG methylation during spermatogenesis than during oogenesis and/or other DNA methyltransferases such as DNMT3C[32] and DNMT3B[33,34] might contribute more to the CG

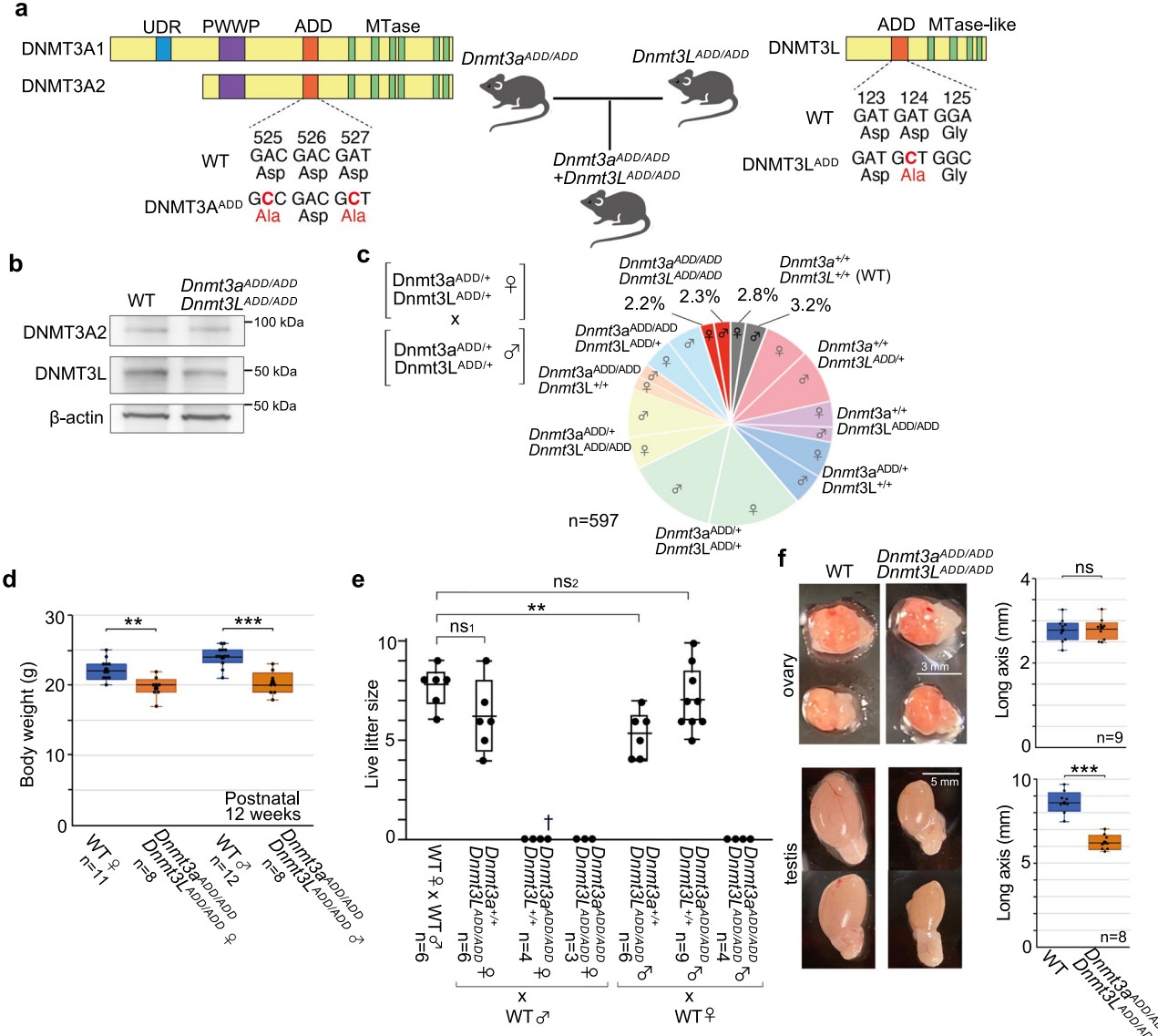

**Fig. 1 | Generation and phenotype of mice carrying ADD mutations in DNMT3A and DNMT3L. a** Structure of mouse DNMT3A isoforms and DNMT3L and positions of the nucleotide substitutions and resulting amino acid substitutions. The introduced nucleotides and resulting amino acids are shown in red. The ADD, PWWP, UDR, and methyltransferase (MTase) motifs of the catalytic domain are indicated by colored boxes. While both DNMT3A1 and DNMT3A2 are expressed in male and female germ cells, DNMT3A2 is the predominant form in FGOs. **b** Western blotting of DNMT3A^ADD and DNMT3L^ADD in wild-type and [$Dnmt3a^{ADD/ADD}$, $Dnmt3L^{ADD/ADD}$] whole testes. This experiment was repeated twice independently. **c** Pie graph showing the genotypes of mice generated by crossing [$Dnmt3a^{ADD/+}$, $Dnmt3L^{ADD/+}$] males and females. A total of 597 mice were genotyped at 3 weeks old. The expected Mendelian ratios for wild-type and [$Dnmt3a^{ADD/ADD}$, $Dnmt3L^{ADD/ADD}$] female and male are respectively 3.12%. **d** Boxplots comparing the body weights of wild-type and [$Dnmt3a^{ADD/ADD}$, $Dnmt3L^{ADD/ADD}$] mice at 12 weeks old. Females and males

were examined separately. All boxplots hereafter are defined as following: central bar, median; lower and upper box limits, 25th and 75th percentiles, respectively; whiskers, minimum and maximum value within the rage of (1st quartile − 1.5*(3rd quartile − 1st quartile)) to (3rd quartile + 1.5*(3rd quartile − 1st quartile)). **p value = 2.0E-03 and ***p value = 4.4E−05 by two-tailed t-test. **e** Fertility of wild-type, $Dnmt3a^{ADD/ADD}$, $Dnmt3L^{ADD/ADD}$, and [$Dnmt3a^{ADD/ADD}$, $Dnmt3L^{ADD/ADD}$] females and males that were crossed with a C57BL/6L partner. One to three stillborn pups were obtained from $Dnmt3a^{ADD/ADD}$ females per delivery, as indicated by a cross (†). ns1 and ns2 p value = 0.10 and 0.47, respectively, and **p value = 4.92E−03 by two-tailed t-test. **f** Representative images of ovaries and testes obtained from wild-type and [$Dnmt3a^{ADD/ADD}$, $Dnmt3L^{ADD/ADD}$] mice. Boxplots on the right show the size (long axis) of these reproductive tissues. ns p value = 0.61 and ***p value = 6.65E-05 by two-tailed t-test. Source data are provided as a Source Data file.

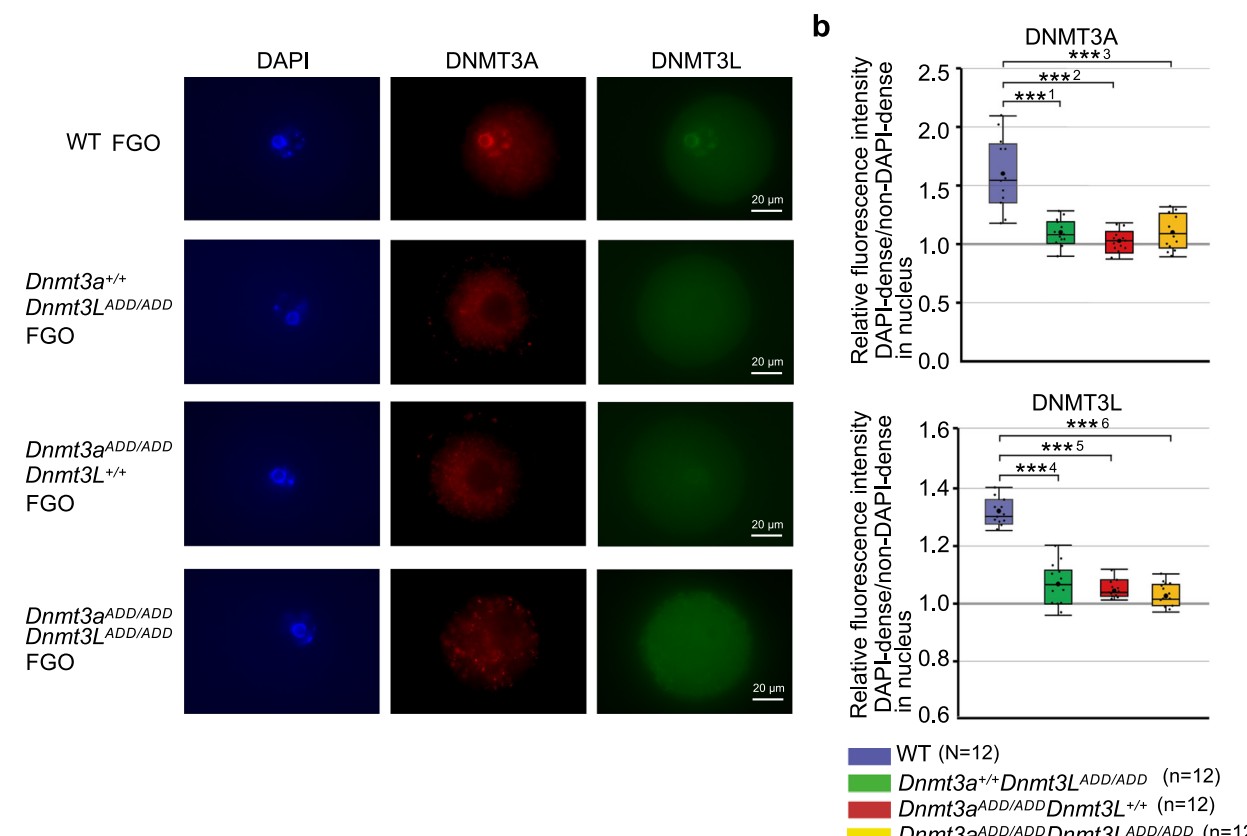

**Fig. 2 | Immunofluorescence staining of FGOs for DNMT3A and DNMT3L carrying ADD mutations. a** Immunofluorescence staining of wild-type, *Dnmt3L*[ADD/ADD], *Dnmt3a*[ADD/ADD], and [*Dnmt3a*[ADD/ADD], *Dnmt3L*[ADD/ADD]] FGOs for DNMT3A (red) and DNMT3L (green). Nuclei were counterstained with DAPI. **b** Boxplots showing the relative fluorescence intensity of DNMT3A (top) and DNMT3L (bottom) in the nucleus of FGOs of indicated genotypes. The relative values were obtained by dividing the fluorescence intensity of the DAPI-dense region by that of the non-DAPI-dense region in the same nucleus. ***1 – ***6 $p$ value = 2.25E-05, 3.10E−06, 5.86E−05, 1.38E-09, 2.23E−13, and 2.13E−13, respectively, by two-tailed t-test. Source data are provided as a Source Data file.

methylation in spermatozoa. Previous studies reported that the severe spermatogenesis defects in DNMT3A and DNMT3L knockout males would link to misactivation of cis-regulatory elements mediated by DNMT3A-dependent DNA methylation in spermatogonial stem cells and retrotransposons methylated dependent on DNMT3L in spermatocytes, respectively[35–37]. However, the double mutant spermatozoa could achieve moderate CG methylation levels at these elements (Fig. 4e, Supplementary Fig. 4b, c). Consistent with this, a certain fraction of [*Dnmt3a*[ADD/ADD], *Dnmt3L*[ADD/ADD]] male germ cells developed beyond elongated spermatid stage, unlike DNMT3A or DNMT3L knockout germ cells[35,36], as described above (Supplementary Fig. 1d, e).

**Transcriptomic profiles of the double mutant FGOs and their subsequent embryos**

To address the impact of the reduced CG methylation caused by the ADD mutations on transcription, we next performed single-cell RNA sequencing (RNA-seq) on *Dnmt3a*[ADD/AD], *Dnmt3L*[ADD/ADD], and [*Dnmt3a*[ADD/ADD], *Dnmt3L*[ADD/ADD]] FGOs, which detected a few hundred differentially expressed genes and a small number of differentially expressed repeat subfamilies (Fig. 5a, Supplementary Fig. 5a, b, Supplementary Data 3, 4). Interestingly, down-regulated genes dominated the differentially expressed genes in all mutants, and certain fractions of them overlapped each other (odds ratio: 6.3–20.0) (Supplementary Fig. 5c, d). Upon multidimensional scaling using single-cell transcriptomic profiles, however, the three types of mutant FGOs and wild-type FGOs were not clearly separated from each other, except for [*Dnmt3a*[ADD/ADD], *Dnmt3L*[ADD/ADD]] FGOs, which formed a loose cluster not intermingling with the other mutant and wild-type FGOs (Fig. 5b). In contrast, wild-type MII oocytes

formed a distinct cluster, suggesting that the drastic loss of CG methylation caused by the ADD mutations had a smaller impact on transcription compared to the transcriptional alterations during oocyte maturation from FGO to MII oocyte.

Given that the three types of mutant FGOs exhibited not-so-different transcriptomic profiles, but clearly different CG methylation levels, we addressed the phenotype of embryos derived by fertilization of oocytes of each mutant type with wild-type spermatozoa. As described above, both *Dnmt3a*[ADD/ADD] and [*Dnmt3a*[ADD/ADD], *Dnmt3L*[ADD/ADD]] females did not deliver viable pups, with *Dnmt3a*[ADD/ADD] females occasionally giving only stillborn pups. We performed in vitro fertilization of [*Dnmt3a*[ADD/ADD], *Dnmt3L*[ADD/ADD]] oocytes with wild-type spermatozoa from JF1 males and transferred the resulting two-cell embryos to the oviducts of pseudo-pregnant females[38]. The JF1 spermatozoa was used to evaluate allele-specific expression in embryos by exploiting the single nucleotide polymorphisms between the JF1 and C57BL/6 J strains (the mutant females were basically C57BL/6J genetic background). While *Dnmt3a*[ADD/ADD] oocytes resulted in embryos with various developmental phenotypes including apparently healthy-looking embryos, as we reported recently[26], all [*Dnmt3a*[ADD/ADD], *Dnmt3L*[ADD/ADD]] oocytes gave embryos with severer defects already at around embryonic day 9.5 ($N = 12$, two transfers) which is reminiscent of embryos derived from *Dnmt3a* or *Dnmt3L* knockout oocytes with global reduction of CG methylation[2,3] (Fig. 5c, Supplementary Fig. 6a, b). As expected from the CG methylation loss in the maternal genome including the ICRs, many maternally imprinted genes were misregulated in the embryos derived from [*Dnmt3a*[ADD/ADD], *Dnmt3L*[ADD/ADD]] oocytes (Fig. 5d, Supplementary Fig. 6c). This is consistent with the crucial role of the imprinted genes in

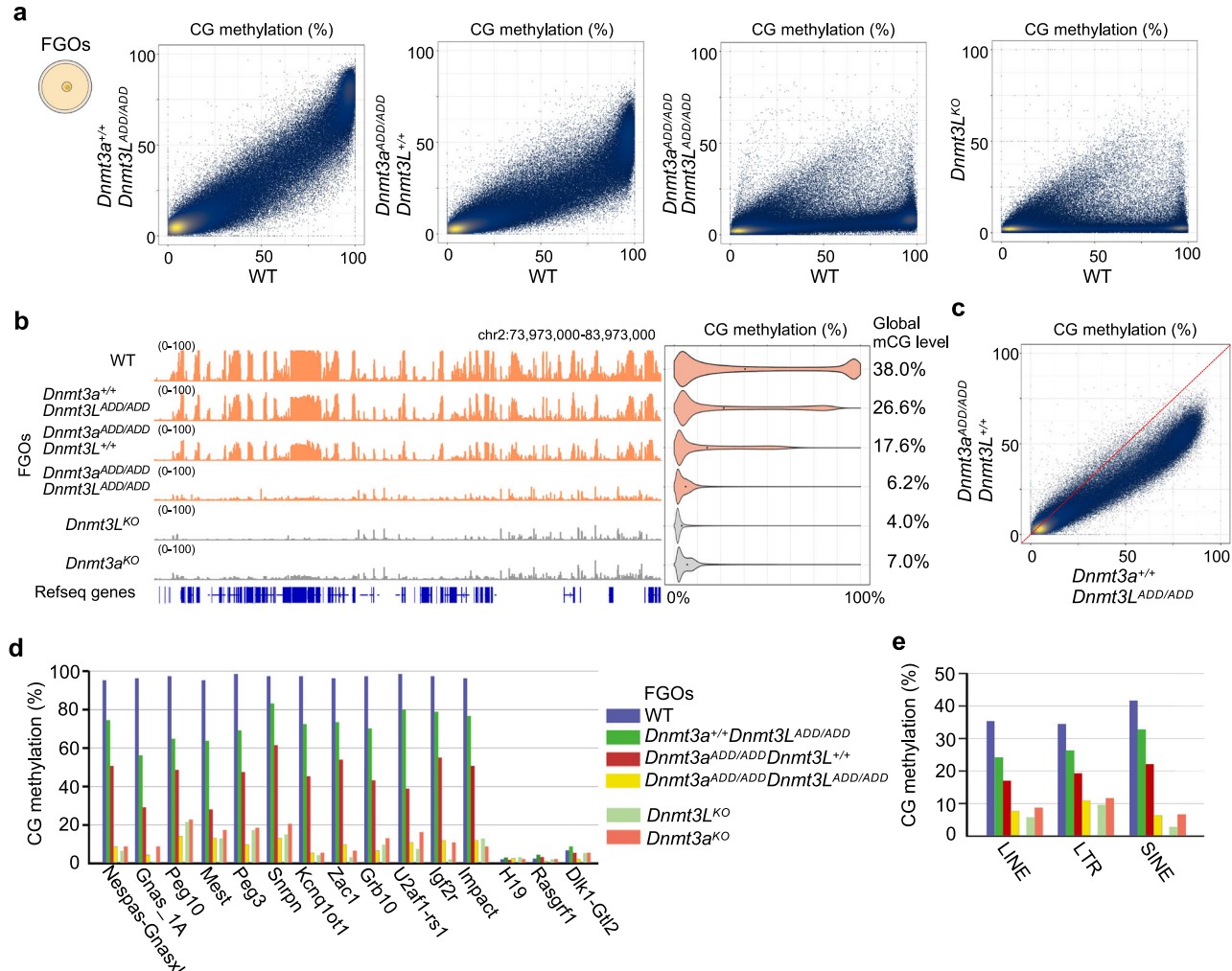

**Fig. 3 | Impact of the ADD mutations in DNMT3A and/or DNMT3L on CG methylation in oocytes. a** Scatter plots comparing CG methylation levels of 10-kb genomic bins between FGOs of the indicated genotypes. A comparison between wild-type and *Dnmt3L* knockout FGOs[30] is also shown. Data from biological replicates were combined after confirming their consistency. **b** Genome browser view of CG methylation levels of 10-kb bins across a 10-Mb region in FGOs of the indicated genotypes. Violin plots on the right show distributions of CG methylation levels of 10-kb bins from the whole genome. The numbers on the right indicate the global CG methylation levels. **c** Scatter plots comparing CG methylation levels of 10-kb genomic bins between *Dnmt3a^ADD/ADD^* and *Dnmt3L^ADD/ADD^* FGOs. **d** CG methylation levels of the maternally and paternally methylated ICRs in FGOs of the indicated genotypes. **e** CG methylation levels of repeat elements (LINEs, LTRs, and SINEs) in FGOs of the indicated genotypes. The color code is as Fig. 3d. Source data are provided as a Source Data file.

---

normal development[1–5], which would explain the severe developmental defects.

## Disproportionately highly accumulated non-CG methylation in the double mutant germ cells

The DNTM3A-DNMT3L complex mediates methylation at both CG and non-CG sites, but subsequent maintenance of the methylated cytosine by DNMT1 and its cofactors occurs only at CG sites during DNA replication, resulting in very low non-CG methylation levels in dividing cells[39–41]. As already known, wild-type FGOs had higher levels of non-CG methylation than wild-type spermatozoa (FGOs, 3.63%; spermatozoa, 0.77%, average of 10-kb bins) (Fig. 6a, b, Supplementary Fig. 7a), likely due to the lack of DNA replication during the oocyte growth phase. However, *Dnmt3a^ADD/ADD^* and *Dnmt3L^ADD/ADD^* FGOs had severely reduced global non-CG methylation levels of 1.17% and 0.67%, respectively, which were approximately one-third to one-fifth of the wild-type level (Fig. 6a, Supplementary Fig. 7a), although the reduction of CG methylation was relatively mild, as described above. Surprisingly, however, when both DNMT3A and DNMT3L had an ADD mutation, the non-CG methylation level became higher than that in the

single mutant FGOs, and the same trend was observed in the double mutant spermatozoa (Fig. 6a–d, Supplementary Data 5). Furthermore, there were thousands of genomic regions that had a non-CG methylation level even higher than that of the wild-type FGOs and spermatozoa (Fig. 6a–d, Supplementary Data 5). These regions corresponded to those showing higher CG methylation levels in double mutant FGOs, which were identified in Fig. 3a (see above) (Supplementary Fig. 7b–d). In total, there were 2,468 10-kb bins that had a 5% higher non-CG methylation level in [*Dnmt3a^ADD/ADD^*, *Dnmt3L^ADD/ADD^*] FGOs than in the wild-type FGOs, and there were 1,261 such bins in [*Dnmt3a^ADD/ADD^*, *Dnmt3L^ADD/ADD^*] spermatozoa. The bisulfite conversion error rates were equally low (0.4–0.9%) in all samples, confirming that the observed non-CG methylation was not an experimental artefact (Supplementary Data 2).

We then addressed the features of the regions showing the high non-CG methylation in the [*Dnmt3a^ADD/ADD^*, *Dnmt3L^ADD/ADD^*] germ cells. Any of the histone marks that we analyzed, including H3K36me2/3[42,43], were not enriched in these regions (Fig. 6c), suggesting that the PWWP domain of DNMT3A, which recognizes histone H3K36me2/3[9–14], does not play a dominant role in chromatin binding even in the absence of

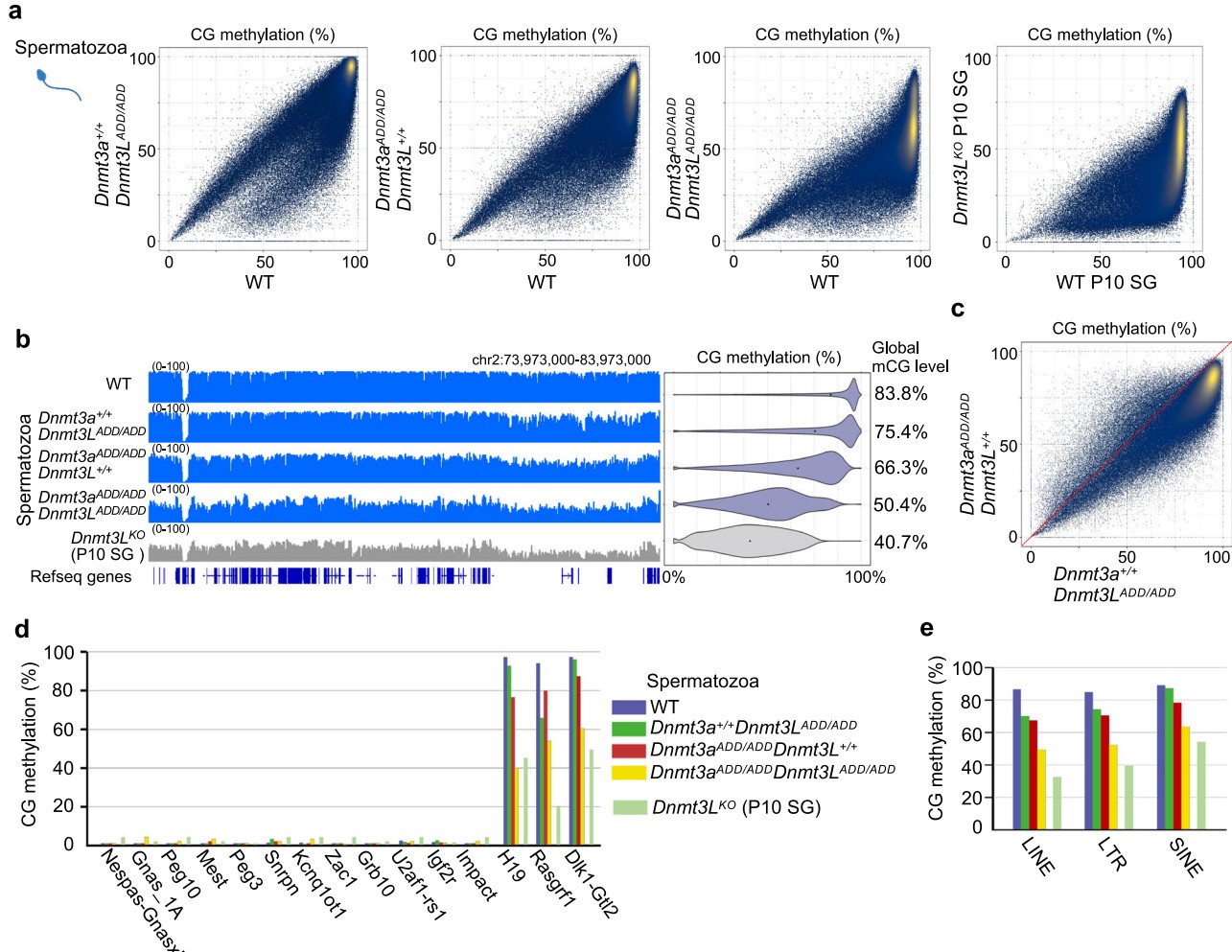

**Fig. 4 | Impact of the ADD mutations in DNMT3A and/or DNMT3L on CG methylation in spermatozoa. a** Scatter plots comparing CG methylation levels of 10-kb bins between spermatozoa of the indicated genotypes. A comparison between wild-type and *Dnmt3L* knockout spermatogonia at postnatal day 10 (P10 SG)[32] is also shown. Data from biological replicates were combined after confirming their consistency. While the wild-type data was produced from one sample, it was consistent with our previous dataset[47]. **b** Genome browser view of CG methylation levels of 10-kb bins across a 10-Mb region (the same region as in Fig. 3b) in spermatozoa of the indicated genotypes. Violin plots on the right show distributions of CG methylation levels in 10-kb bins from the whole genome. The numbers on the right indicate the global CG methylation levels. **c** Scatter plots comparing CG methylation levels of 10-kb genomic bins between *Dnmt3a^ADD/ADD* and *Dnmt3L^ADD/ADD* spermatozoa. **d** CG methylation levels of the maternally and paternally methylated ICRs in spermatozoa of the indicated genotypes. **e**, CG methylation levels of repeat elements (LINEs, LTRs, and SINEs) in spermatozoa of the indicated genotypes. The color code is as Fig. 4d. Source data are provided as a Source Data file.

the ADD-H3K4me0 interaction. In addition, preferential methylation of the CA sequence among the three non-CG sequences (CA, CC, and CT) in wild-type FGOs[30] was compromised in the double mutant cells (Fig. 6e), and there was no clear correlation between the high non-CG methylation and transcription (Supplementary Fig. 7e). Interestingly, the regions with high non-CG methylation were often shared between [*Dnmt3a^ADD/ADD*, *Dnmt3L^ADD/ADD*] FGOs and spermatozoa (966 of the 1261 bins (76.6%) in [*Dnmt3a^ADD/ADD*, *Dnmt3L^ADD/ADD*] spermatozoa) (Supplementary Fig. 7f), and no specific motifs other than (CA)n repeat were enriched in these regions (the CA density itself was comparable between the high non-CG methylation regions and randomly selected regions, 7.55% versus 7.44%) (Fig. 6f, Supplementary Fig. 7g). These findings suggest that factors common to the cell types, such as the genomic sequences themselves or their local DNA structures, may have a certain affinity for the DNMT3A-DNMT3L complex regardless of the ADD-H3K4me0 interaction. Consistent with this idea, wild-type germ cells that would have sufficient DNMT3A-DNMT3L complexes in the nucleus (Fig. 2) also showed relatively higher non-CG methylation levels at these regions (Fig. 6g). Taken together, while a complete loss of the ADD-H3K4me0 interaction leads to a drastic reduction of CG

methylation across the genome, non-CG methylation is highly accumulated in specific regions in both oocytes and spermatozoa.

## Discussion

Our study revealed the combined and differential roles of the DNMT3A-ADD and DNMT3L-ADD domains in the establishment of DNA methylation landscapes in both female and male germ cells. While a functional loss of the ADD domain of either DNMT3A or DNMT3L leads to a moderate CG-methylation reduction, a combined loss leads to much more severe reduction in the global CG methylation level, especially in oocytes, which is similar to the impact of the Dnmt3a/Dnmt3L knockout mutations[30,32]. Such a severe CG-methylation loss also occurs at the maternally methylated ICRs and, probably due to the misregulation of the linked imprinted genes, double mutant oocytes cannot support normal development. Given that DNMT3A^ADD still has a PWWP domain that interacts with histone H3K36me2/3[9–14], which normally coexists with H3K4me0[13,42], one could imagine that the DNMT3A^ADD-DNMT3L^ADD complex would still interact with chromatin in double homozygous cells. However, a loss of CG methylation occurs throughout the genome in double mutant oocytes, regardless of

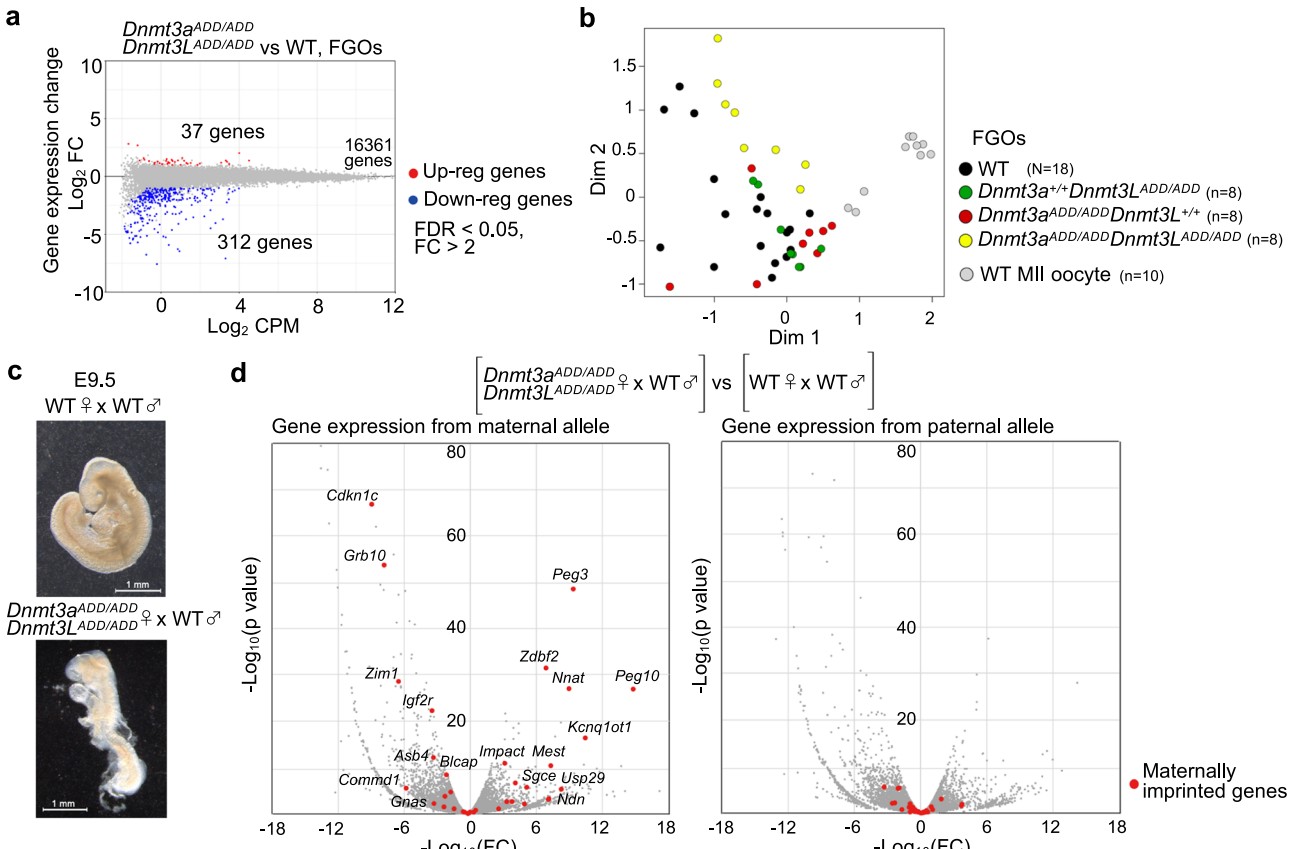

**Fig. 5 | Transcriptome analysis of the double mutant FGOs and subsequent embryos. a** Changes in gene expression detected between wild-type and [*Dnmt3a^{ADD/ADD}*, *Dnmt3L^{ADD/ADD}*] FGOs. Genes that are up-regulated and down-regulated in [*Dnmt3a^{ADD/ADD}*, *Dnmt3L^{ADD/ADD}*] FGOs are plotted in red and blue, respectively (fold-change > 2, FDR < 0.05). **b** Multi-dimensional scaling plots of the gene expression profiles of FGOs of the indicated genotypes. The profiles of the wild-type MII oocytes are also shown. Each color-coded dot shows single-cell data. **c** Representative images of embryos recovered at E9.5. Wild-type embryo (top) and

embryo obtained from [*Dnmt3a^{ADD/ADD}*, *Dnmt3L^{ADD/ADD}*] oocyte fertilized with wild-type spermatozoa (bottom). This experiment was repeated twice independently. **d** Volcano plots showing gene expression differences of each parental allele between embryos obtained from wild-type and [*Dnmt3a^{ADD/ADD}*, *Dnmt3L^{ADD/ADD}*] oocytes. Data from the maternal (left) and paternal allele (right) are separately shown. Red dots indicate the maternally imprinted genes (*n* = 35). P values by the exact test under a negative binomial distribution. Source data are provided as a Source Data file.

presence/absence of H3K36me2/3 marks. These results suggest that the PWWP-H3K36me2/3 interaction cannot compensate for the absence of ADD-H3K4me0 interaction, in terms of the efficiency of the CG methylation activity. Given that ADD-H3K4me0 binding changes the structure of DNMT3A and releases its own methyltransferase domain[20], a loss of the ADD-H3K4me0 interaction could also affect the PWWP-H3K36me2/3 interaction. In addition, DNMT3A interacts with other proteins such as MYC and MECP2, and the binding of MECP2 with the DNMT3A-ADD domain affects the autoinhibitory conformation of DNMT3A[44–46]. It would be worth studying such structural changes in the DNMT3A-DNMT3L heterotetramer with or without ADD-H3K4me0 binding.

Consistent with the global reduction in CG methylation, we observed significant decrease in accumulation of DNMT3A^{ADD} and DNMT3L^{ADD} proteins in the nucleus of FGOs. Notably, the ADD mutation of either protein affected the nuclear accumulation of both proteins. In addition, the degree of reduction of immunofluorescence signal was almost equivalent to that observed in [*Dnmt3a^{ADD/ADD}*, *Dnmt3L^{ADD/ADD}*] FGOs, although the degrees of CG methylation loss were significantly different as described above. Because immunofluorescence signal in the nucleus is not a direct measure of protein-chromatin interaction, a highly sensitive method to quantify chromatin-bound proteins would be required to resolve the discrepancy.

While our study uncovered the important roles of the ADD domains in both oocytes and spermatozoa, we also observed different

phenotypes in the female and male gametogenesis. First, while [*Dnmt3a^{ADD/ADD}*, *Dnmt3L^{ADD/ADD}*] females produced a normal number of FGOs, double mutant males showed reduced testis size and impaired spermatogenesis. Yet, the global CG methylation level of the double mutant spermatozoa (50.4%) was clearly higher than that of the double mutant oocytes (6.2%). The double mutant males showed reduced number of spermatids in the seminiferous tubules and severe impairment in motility and viability of spermatozoa in the epididymis, but this phenotype was less severe than that of *Dnmt3a* or *Dnmt3L* knockout males[35,36]. Since de novo DNA methylation appears to regulate several distinct processes of spermatogenesis[35,36,47], it would be interesting to investigate which spermatogenic stage is more dependent on the ADD domains for establishment of the proper DNA methylation landscape. Such a study will help us understand why the double mutant male germ cells can develop beyond the elongated spermatid stage but show impaired motility and viability.

Lastly, while non-CG methylation globally disappeared in *Dnmt3a^{ADD/ADD}* and *Dnmt3L^{ADD/ADD}* oocytes and spermatozoa, we found thousands of genomic regions with high non-CG methylation accumulation in [*Dnmt3a^{ADD/ADD}*, *Dnmt3L^{ADD/ADD}*] germ cells. Although we did not observe any specific histone marks or transcription factor binding motifs enriched at these regions, a majority of such regions were shared between oocytes and spermatozoa, suggesting that a factor common to the two cell-types, such as the genomic sequences themselves, attracts the DNMT3A-DNMT3L complex. One slightly enriched

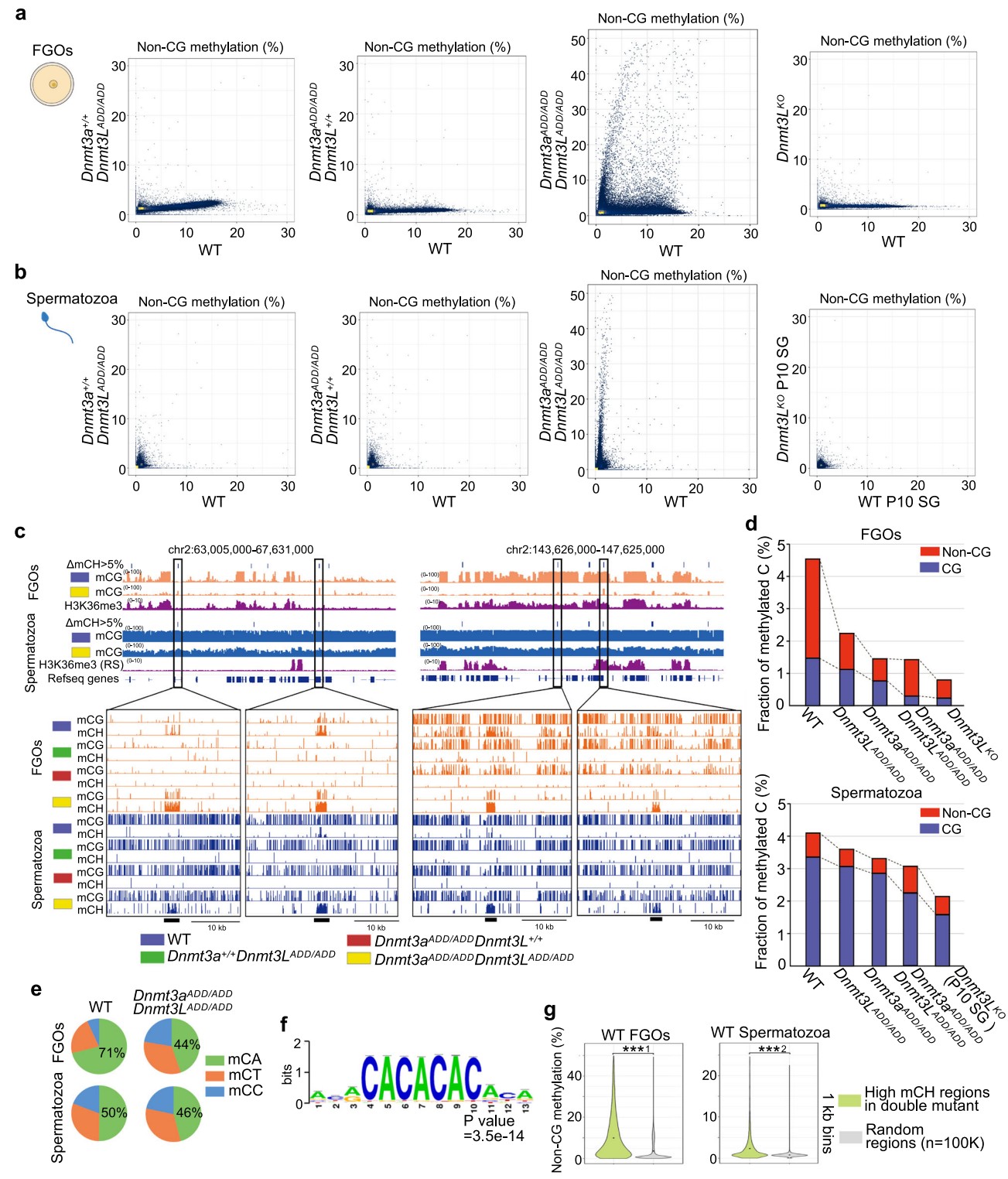

sequence that we identified was (CA)n repeat, but we do not know how it attracts the enzyme complex. It is also possible that specific structural patterns of genomic DNA have a certain affinity for the complex. In any case, we speculate that such an interaction is normally weak and becomes evident only when the ADD-H3K4me0 interaction is absent. The underlying mechanism of such non-CG methylation warrants further investigation.

Our study uncovered the roles of the DNMT3A-ADD and DNMT3L-ADD domains and their combined impact in germ cells and presented an additional case for the high accumulation of non-CG methylation in vivo. DNMT3A and DNMT3L have been recognized as the main players in de novo DNA methylation and their mutations, including those found in the ADD domains, have been implicated in various human diseases[21–24]. Thus, our findings provide additional insights into the

**Fig. 6 | Accumulation of non-CG methylation in oocytes and spermatozoa of double mutant mice. a** Scatter plots comparing non-CG methylation levels of 10-kb genomic bins between FGOs of the indicated genotypes. A comparison between wild-type and *Dnmt3L* knockout FGOs[30] is also shown. **b** Scatter plots comparing the non-CG methylation levels of 10-kb bins between spermatozoa of the indicated genotypes. A comparison between wild-type and Dnmt3L knockout P10 SG[32] is also shown. **c** Genome browser view of CG methylation (mCG) and non-CG methylation (mCH) levels in FGOs and spermatozoa of the indicated genotypes. Two genomic regions of 4.6 Mb (left) and 4.0 Mb (right) are shown. H3K36me3 ChIP-seq peaks in wild-type FGOs and round spermatids (RS) are also shown. The 10-kb bins exhibiting non-CG methylation levels > 5% higher in [*Dnmt3a*$^{ADD/ADD}$, *Dnmt3L*$^{ADD/ADD}$] oocytes and spermatozoa compared to their wild-type counterparts are indicated in a row designated "ΔmCH>5%". Four representative regions containing such high

mCH bins are indicated by open boxes and their zoomed-in snapshots are displayed at the bottom. **d** Bar graphs showing the fractions of methylated cytosines in all cytosines for FGOs (top) and spermatozoa (bottom) of the indicated genotypes. Each bar is divided into subsegments representing mCH and mCG. **e** Pie charts showing the ratios of methylated CA, CT, and CC in FGOs (top) and spermatozoa (bottom) of the indicated genotypes. **f** A motif showing enrichment in high-mCH 1-kb bins of [*Dnmt3a*$^{ADD/ADD}$, *Dnmt3L*$^{ADD/ADD}$] FGOs and spermatozoa ($N = 1855$) over randomly selected 1-kb bins. **g** Violin plots showing distributions of non-CG methylation levels of 1-kb bins in wild-type FGOs (left) and spermatozoa (right). The 1-kb bins exhibiting non-CG methylation levels > 20% higher in [*Dnmt3a*$^{ADD/ADD}$, *Dnmt3L*$^{ADD/ADD}$] compared to their wild-type counterparts were compared with randomly selected bins. ***[1] and ***[2] $p$ value = 5.92E-291, and 3.44E−72, respectively, by two-tailed t-test. Source data are provided as a Source Data file.

molecular mechanisms of de novo DNA methylation and should contribute to our understanding of these related diseases.

## Methods

### Mouse husbandry
All animal experiments were performed according to the ethical guidelines of Kyushu University and the protocols were approved by the Institutional Animal Care and Use Committee of Kyushu University (A22-087-1). Mice were group-housed in a specific-pathogen-free facility under standard housing conditions (12-h light/dark cycle, temperature 20–22 °C, humidity 40-60%, and free access to water and food).

### Mutant mice
*Dnmt3L*$^{ADD}$ mutant mice were generated using CRISPR-Cas9[48]. The pX330 plasmid encoding Cas9 and guide RNA (gRNA) and single-stranded donor oligonucleotides (ssODNs) were microinjected into mouse zygotes derived from (C57BL/6N × C3H/HeN) F1 females crossed with C57BL/6J males. The injected zygotes were transferred to the oviducts of pseudo-pregnant ICR females. *Dnmt3a*$^{ADD}$ mutant mice were also generated in the same way in our previous study[26]. The pups were genotyped using PCR-based Sanger sequencing. The gRNA and ssODN sequences and primers used for genotyping are listed in Supplementary Data 1. A male carrying the expected *Dnmt3a*$^{ADD}$ or *Dnmt3L*$^{ADD}$ mutation was crossed with C57BL/6J females, and offspring carrying the mutations were backcrossed to C57BL/6J at least five times. Heterozygous *Dnmt3a*$^{ADD/+}$ and *Dnmt3L*$^{ADD/+}$ mice were intercrossed to generate double heterozygous *Dnmt3a*$^{ADD/+}$*Dnmt3L*$^{ADD/+}$ mice. [*Dnmt3a*$^{ADD/ADD}$, *Dnmt3L*$^{ADD/ADD}$] mice were generated by crossing *Dnmt3a*$^{ADD/+}$*Dnmt3L*$^{ADD/+}$ males and females.

### Fertility testing
C57BL/6J, *Dnmt3a*$^{ADD/ADD}$, *Dnmt3L*$^{ADD/ADD}$, and [*Dnmt3a*$^{ADD/ADD}$, *Dnmt3L*$^{ADD/ADD}$] females and males of 8 weeks or older were placed with a C57BL/6J partner in the same cage, and vaginal plugs were checked the next morning. The number of viable and stillborn pups were counted.

### Oocyte and spermatozoa collection
FGOs were harvested from adult ovaries (10–12 weeks old) by needle puncture, and cumulus cells were removed by capillary washing. FGOs were pooled in M2 medium (Sigma) and stored at −80 °C. Spermatozoa was released from the cauda epididymis of adult male mice and collected using the swim-up method. In the case of [*Dnmt3a*$^{ADD/ADD}$, *Dnmt3L*$^{ADD/ADD}$] spermatozoa, viable spermatozoa were selected by mouth pipetting.

### IVF, two-cell embryo transfer, and embryo dissection
Before obtaining MII oocytes, inhibin antiserum (CARD HyperOva, Kyudo) and human chorionic gonadotropin were sequentially injected into 11–18-week-old females. Cumulus-oocyte complexes were

collected from the ampulla of oviducts and fertilized with JF1 spermatozoa in modified human tubal fluid (CARD mHTF, Kyudo). Cumulus cells were removed by capillary washing, and fertilized eggs were cultured in EmbryoMax KSOM Medium (1X) w/ 1/2 Amino Acids (Merck Millipore) at 37 °C and 5% $CO_2$. After overnight culture, two-cell embryos were transferred into the oviducts of pseudo-pregnant ICR females. Embryos and placentas were dissected from the uteri of the pregnant females on embryonic day 9.5.

### Immunofluorescence staining
Immunofluorescence staining was performed to visualize the subcellular localization of DNMT3A and DNMT3L in the cells. Briefly, the FGOs were fixed with 4% paraformaldehyde for 20 minutes at room temperature. FGOs were then permeabilized with 0.1% Triton X-100 for 10 minutes and blocked with 5% BSA for 1 hour at room temperature. The FGOs were then incubated with primary antibodies against DNMT3A (NOVUS, 64B14446) or DNMT3L (Abcam, ab194094) (dilution 1:500) overnight at 4 °C. After overnight incubation, FGOs were incubated with anti-rabbit IgG (H + L) CF 488 A and anti-mouse IgG (H + L) CF 594 secondary antibodies (Biotium) (dilution 1:1000) for 1 hour at room temperature. The nuclei were counterstained with DAPI. Immunostained FGOs were observed using a fluorescence microscope (KEYENCE, BZ-X800).

### Western blotting
Testicular tissue obtained from wild-type and [*Dnmt3a*$^{ADD/ADD}$, *Dnmt3L*$^{ADD/ADD}$] males of 10 weeks old was homogenized in ice-cold lysis buffer (50 mM Tris-HCl, 150 mM NaCl, 1% Triton X-100, 0.5% Sodium Deoxycholate, 0.1% SDS, 1 mM EDTA, 1x protease inhibitor). Proteins were subjected to 8 or 10% SDS-polyacrylamide gel electrophoresis (SDS-PAGE) for 60 minutes at 100 V and transferred to a PVDF membrane. The membrane was blocked with skimmed milk for 30 minutes at room temperature and incubated overnight at 4 °C with primary antibodies against DNMT3A (NOVUS, 64B1446), DNMT3L (Abcam, ab194094) (dilution 1:1000), and β-actin (Santa Cruz sc-69879) (dilution 1:1000) in blocking buffer. Horseradish peroxidase (HRP)-conjugated anti-mouse IgG and anti-rabbit IgG antibodies (Abcam, ab6789 and ab6721) (dilution 1:30,000) were used as the secondary antibodies. The membrane was incubated with Chemi-Lumi One Ultra (Nacalai Tesque) and the resulting chemiluminescence signals were detected using an ImageQuant LAS3000 mini (Cytiva).

### WGBS
We used 100 wild-type and *Dnmt3L*$^{ADD/ADD}$ FGOs and 48 [*Dnmt3a*$^{ADD/ADD}$, *Dnmt3L*$^{ADD/ADD}$] FGOs for library preparation. Samples were directly spiked with 5 pg of unmethylated lambda phage DNA (Promega) and subjected to bisulfite treatment and library construction using the post-bisulfite adaptor tagging (PBAT) method[49]. To prepare spermatozoa WGBS libraries, we used approximately 5000 wild-type, *Dnmt3a*$^{ADD/ADD}$, and *Dnmt3L*$^{ADD/ADD}$ spermatozoa and 500 [*Dnmt3a*$^{ADD/ADD}$, *Dnmt3L*$^{ADD/ADD}$] spermatozoa. Spermatozoa were incubated overnight

in 25 μL lysis buffer (100 mM Tris·Cl (pH 8.0), 10 mM EDTA, 500 mM NaCl, 1% sodium dodecyl sulfate (SDS), and 2% β-mercaptoethanol) and 2 μL of 20 μg/μL proteinase K at 55 °C. After overnight incubation, the samples were directly spiked with 50 pg of unmethylated lambda phage DNA (Promega), followed by bisulfite treatment and library construction as described for the FGO libraries. WGBS libraries from FGOs and spermatozoa were amplified using KAPA library amplification kit (KAPA) for five cycles. Library concentrations were determined using the KAPA Library Quantification Kit (KAPA). WGBS libraries were sequenced on a NovaSeq 6000 (Illumina) using a NovaSeq 6000 SP Reagent Kit (Illumina) to generate 108-nucleotide single-end reads. Two biological replicates were generated for each sample. The reads were mapped to mouse genome mm10 using Bismark v0.20.0[50], and Bowtie v1.3.1[51]. Data on CG and non-CG methylation with genomic coverage ranging from 5 to 500 times were used for subsequent analyses. Bedtools v2.25.0[52] was used for comparing genomic features.

### RNA-seq
Each single-cell RNA-seq library was generated from a single FGO or a single MII oocyte using SMART-Seq HT Kit (Takara) according to the manufacturer's instructions. For embryos recovered at embryonic day 9.5, total RNA was extracted using AllPrep DNA/RNA Mini kit (QIAGEN) and RNA-seq libraries were generated using NEBNext polyA isolation module (NEB 7490S) and NEBNext Ultra II Directional RNA Library Prep Kit for Illumina (NEB 7760) according to the manufacturer's instructions. RNA-seq libraries were subjected to paired-end sequencing (53 nucleotides each) on NovaSeq 6000 (Illumina) using a NovaSeq 6000 SP Reagent Kit (Illumina). Eight to ten biological replicates were generated for each sample. RNA-seq reads (paired-end, 100 bases) were aligned against the mouse mm10 genome assembly using STAR v2.5.3a[53]. The mapped reads were counted using featureCounts v1.5.3[54], HTSeq 2.0.5[55], and the output files from multiple replicates were subsequently analyzed using edgeR v3.14.0[56] to estimate the transcript abundance and to detect the differentially expressed genes (FDR < 0.05, fold change > 2). RPKM was calculated using an in-house pipeline. GO analysis was performed using DAVID v2022q1[57].

### Motif analysis
Enrichment analysis of known DNA binding motifs was performed using Regulatory Sequence Analysis Tools (RSAT)[58]. Sequences of 1-kb bins exhibiting non-CG methylation levels > 20% higher in [*Dnmt3a$^{ADD/ADD}$*, *Dnmt3L$^{ADD/ADD}$*] compared to their wild-type counterparts were analyzed with default parameters and 10,000 of randomly selected 1-kb sequences were used as background.

### Reporting summary
Further information on research design is available in the Nature Portfolio Reporting Summary linked to this article.

## Data availability
All sequencing datasets of WGBS and RNA-seq generated in this study have been deposited in the Gene Expression Omnibus (GEO) under accession code GSE238228. The WGBS data from *Dnmt3a$^{ADD/ADD}$* FGO is available in the Sequence Read Archive database (accession code PRJDB12492). The WGBS data of *Dnmt3a* and *Dnmt3L* knockout FGOs and wild-type and Dnmt3L knockout spermatogonia at postnatal day 10 that were used in this study are available in DDBJ (accession code DRX001586 and DRX001588) and GEO (accession code GSE84140), respectively. H3K36me3 ChIP-seq data in FGOs and round spermatogonia that were also used in this study are available in GEO under accession codes GSE183969 and GSE108717, respectively. The mouse reference genome data (mm10) and the SNP information for the JF-1 mouse genome are available from UCSC and RIKEN BRC, respectively. Source data are provided with this paper.

## Code availability
All code used in this study was previously published and no customized code was used in this manuscript.

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

## Acknowledgements

We would like to thank Tomoko Hanagiri, Miho Miyake, Wen Chaoqing, and Tomomi Akinaga, as well as all the other members of the Sasaki laboratory (Kyushu University) for their support and technical assistance. We would like to acknowledge the help of Kanako Ichikawa and Ryo Ugawa (Laboratory for Research Support, Medical Institute of Bioregulation, Kyushu University) for their technical assistance. This work was supported by JSPS KAKENHI Grant Numbers JP22H04675, JP22K15037, and JP24K09414 to N.K., JP22K15125 to K.S., and JP18H05214 to H.S., and also supported by Designated research grants from the Uehara Memorial Foundation to N.K. and Research grant of oncology from MSD Life Science Foundation to N.K.

## Author contributions

N.K. and H.S. conceived and designed the experiments. N.K. R.U., S.U., H.O., and K.S. performed mouse experiments. N.K. performed the library preparation. N.K. performed bioinformatics analysis. N.K. and H.S. interpreted the data. All authors contributed to the writing of the manuscript.

## Competing interests

The authors declare no competing interests.
