## [Peer Review File · Nature Communications]

Combined and differential roles of ADD domains of DNMT3A and DNMT3L on DNA methylation landscapes in mouse germ cellsReviewers' Comments:

Reviewer #1:

Remarks to the Author:

The study addresses the role in vivo of the ADD domains of the de novo DNA methyltransferase DNMT3A and its non-catalytic partner DNMT3L. It does so in the context of male and female germlines in mice: in the female germline, both DNMT3A and DNMT3L are essential for global de novo DNA methylation; in the male germline, both proteins have major roles, but there is more redundancy because of DNMT3B and DNMT3C. The ADD domain is thought to be a major regulator of the functional interaction of DNMT3 proteins with chromatin, as it binds unmethylated lysine 4 of histone H3 (H3K4me0), and this interaction is required to alleviate autoinhibition of catalytic activity. Previous work from this group had generated a DNMT3A-ADD mutant thought to abrogate the interaction with H3K4me0 and evaluated its effect during male and female gametogenesis. In the present study, the authors have generated an analogous DNMT3L-ADD mutant and tested its effect during male and female gametogenesis in isolation and in combination with the DNMT3A-ADD mutant.

In brief, the authors find that:

1. the ADD mutations cause reduced association of DNMT3A and DNMT3L on chromatin in fully-grown oocytes (FGOs), as assessed by immunofluorescence.
2. the ADD mutations cause reduced CG-methylation in FGOs, which is most severe in the double mutants, essentially equivalent to lack of de novo methylation. In spermatogonia, the effects on CG methylation are more modest, probably because of the partial redundancy with DNMT3B and DNMT3C.
3. scRNA-seq analysis of FGOs indicates limited effect on gene expression, with a small number of differentially expressed genes (DEGs) identified, most of which are down-regulated.
4. embryos developing from double-mutant oocytes fail at mid-gestation and display abnormal expression of imprinted genes, consistent with the observed impaired de novo methylation of their germline DMRs in oocytes.
5. non-CG methylation, which is known to be high in FGOs, is also severely affected in the double mutants; however, the authors detect sites of enhanced non-CG methylation (also in spermatogonia), which show enrichment in CA repeats. The authors speculate that these repeats may become sites of recruitment of DNMT3A/3L when the normal recruitment to H3K4me0 is abrogated.

In general, this is a very nicely conducted study, the data are strong and analysed robustly, and the manuscript is presented clearly and succinctly. The authors are able to infer that the ADD domains appears dominant for DNMT3A/3L interaction with and function on chromatin: presence of an intact PWWP domain directed towards H3K36me2/me3 domains is insufficient for efficient de novo methylation in oocytes. On the other hand, other molecular findings are not provided with mechanistic insights.

I have a few minor suggestions:

1. The authors conclude there is no impact of the Dnmt3a/Dnmt3L-double mutant on oocyte development from the normal gross morphology of ovaries and from the ability to isolate comparable numbers of FGOs which are grossly normal in appearance (bright field). A slightly deeper analysis could be considered; for example, checking nuclear maturation rate (proportion of SN versus NSN oocytes). I just have in mind the transcriptome changes in FGOs from all mutant genotypes, and whether characteristics such as nuclear maturation could contribute to transcriptome differences.

2. In some places, the analysis could be deeper to aid mechanistic inferences. For example, the authors do not present much analysis of the FGO scRNA-seq data. There are essentially comparable numbers of DEGs called in the three mutant genotypes (276 in the Dnmt3L-ADD mutant, 421 in the Dnmt3A-ADD mutant, 349 in the double mutant), but in all cases there is an excess of down-regulated genes, suggesting a biological effect rather than false discovery, but for which the authors provide no explanation or speculation. I do not anticipate there will be much useful signal from GO analysis. But it would be useful to know whether there is any overlap amongst the DEGs between the three

genotypes.

It is a general observation that substantial loss of CG-methylation in mouse oocytes has very limited impact on gene transcription, but aberrant gains of methylation could perhaps result in impaired transcription. Therefore, it would be interesting to explore whether there were any associated between down-regulated genes and sites of enhanced non-CG methylation.

Reviewer #2:

Remarks to the Author:

The de novo DNA methyltransferase Dnmt3a and its accessory factor Dnmt3L are both essential in establishing the DNA methylation landscapes in mouse germ cells. In this manuscript, Kubo et al. provided genetic evidence that the ADD domains of Dnmt3a and Dnmt3L have both distinct and redundant functions in this regard. The data are convincing, and the finding is significant in understanding the mechanisms of action by the DNA methylation machinery in germ cells.

I have a few minor suggestions/comments for the authors to consider:

1. Fig. 2a and Extended Data Fig. 2a. The Dnmt3a signal is present in smaller areas in the cytoplasm compared to the Dnmt3L signal, except in double homozygous oocytes. Please comment.

2. Page 6, lines 138-141: For the statement "Notably, when both DNMT3A-ADD and DNMT3L-ADD had mutations, the global CG methylation level was much more reduced ([Dnmt3aADD/ADD, Dnmt3LADD/ADD], 6.2%), and to our surprise, comparable with that of DNMT3A or DNMT3L knockout FGOs (Fig. 3a, b, c, Extended Data Fig. 3a)", Fig. 3c should NOT be cited.

3. Fig. 3a, a small number of CpG sites normally hypomethylated apparently gained methylation in double homozygous FGOs. From subsequent results and description, I'd guess that those sites were in the regions that gained non-CpG methylation. If so, I'd suggest that, when describing Fig. 3, the gain of methylation at a small number of CpG sites be mentioned, and "(see below)" can be added to avoid the impression that the data were ignored. If those CpG sites are not in regions with gain of non-CpG methylation, where are they located, and are there general sequence contexts or features?

4. Page 11-12, lines 273-277: "In addition, the degree of reduction of immunofluorescence signal was almost equivalent to that observed in [Dnmt3aADD/ADD, Dnmt3LADD/ADD] FGOs, although the degrees of CG methylation loss were significantly different as described above. Because immunofluorescence staining is not quite quantitative, a highly sensitive method to quantify chromatin-bound proteins would be required to resolve the discrepancy". The last sentence appears to suggest that the degree of reduction of immunofluorescence signal could be more severe in double homozygous FGOs, but the difference could not be detected by the non-quantitative method. While that could be one possibility, another possibility is that the degree of reduction was indeed equivalent. In that case, I'd like to hear alternative/additional explanations.

Reviewer #3:

Remarks to the Author:

Kubo et al. have systematically investigated the functional role of an intact ADD domain in DNMT3A and DNMT3L in mouse development. One strength of this paper is that the authors did not delete the entire domain, with undefined effects on the functional integrity of the remaining protein parts, but they introduced defined point mutations that disrupt the interaction of the ADD domains with H3-tails. The reported data are interesting and partially surprising. On the more negative side, novelty is compromised by the authors own report of the DNMT3A ADD mutant, which covers 25% of the data included here. Moreover, the paper is largely descriptive and not many mechanistic insights are

provided and some technical and conceptual questions are left unanswered.

Major comments:

- 1) I am a bit concerned that the number of animals used for Fig. 1D is low. This should be increased.
- 2) Statistics should be shown for Fig. 1f to justify the corresponding statements in the text.
- 3) Heterochromatic localization: Literature should be cited for heterochromatic localization of DNMT3A/3L. In this paragraph the term heterochromatin/heterochromatic should be used throughout (lines 112, 116, 118, 120, 124, 127).
- 4) The number of FGO analyzed in Fig. 2 is not sufficient to draw strong conclusions.
- 5) Apparently, the results presented in Fig. 2 do not fit to the rest of the data, as heterochromatic localization was lost for all mutants (including the double mutant) equally, but strongest methylation changes were observed in double mutants. In relation to this there is a conceptual mistake in lines 133 and 246, because Fig. 2 only shows reduced heterochromatic localization but not reduced chromatin binding.
- 6) One critical question regarding Fig. 3 and all the methylation related data is about reproducibility. At least the analysis should be repeated with WT and double mutant and the resulting variance documented.
- 7) Additional experiments should be conducted to determine if other DNMT3 enzymes, most likely DNMT3C, are affecting the data obtained in spermatozoa. At present the data set is inconclusive. Breeding may be performed with heterozygous DNMT3C KO animals.
- 8) In Fig. 5B, the question appears, if the authors conducted MCA without including the MII oocytes as an outgroup. Did the authors test MII oocytes from double mutants?
- 9) The increase in non-CpG methylation in double mutants is one of the most striking and surprising findings of this work. However, many aspects related to this observation remained unclear and I am not convinced by the mechanistic model proposed by the authors to explain this observation. The double mutant WGBS analysis should be repeated (together with internal controls of unmethylated DNA) to exclude incomplete conversion as a reason for the observations. Incomplete conversion would easily explain the enrichment of non-CpG at poly-CA regions, because at these sites, the CpA density is highest, hence the chances to observe a randomly occurring incomplete conversion in CpA context are highest. Alternatively, the poly-CA sequence may be related with conversion problem, perhaps by forming strong secondary structures.
- 10) Given the known specificity of DNMT3A, the non-CpG methylation should be mainly CpA and this should be shown.

Minor comments

- 11) It would be good to mention the interaction of DNMT3A1 via its UDR with H2AK119ub1 in the introduction as well (PMID: 33986537), and show the UDR in Fig. 1A as well.
- 12) Fig. 1C is difficult to read. Perhaps it would help to draw observed/expected ratios, then readers do not need to do the mendelian calculations themselves.
- 13) Where are the regions plotted in Fig. 6F in the WT panels in Fig. 6A and B? Same violin plots as in 6F should be shown for the double mutants.
- 14) There is a conceptual mistake in the discussion (line 264) because the ADD and PWWP domains will target DNMT3A to distinct genomic regions. Hence, PWWP can never "compensate" for the loss of ADD function.
- 15) It should be considered, at least in discussion, that the ADD domain of DNMT3A mediates interaction also with other proteins (not only H3) (see for example: PMID 16322236, 30102379, 16322236, 15616584) and at least in the case of MECP2 the interaction has been mapped to the same interface as the H3-tail interaction (PMID: 36519786). Hence the mutated ADD domains studied by Kubo et al. have not only lost interaction with H3-tails but also with MECP2.

We would like to thank the reviewers for considering our manuscript and providing constructive feedbacks! We have thoroughly revised the manuscript by including new experimental data and computational analyses. We believe that the additional results improve the quality of the manuscripts and further strengthen our conclusions. We also revised the maintext to state the novel findings of our study more precisely and clearly. Below, we summarize the major changes made in the revised manuscript, followed by point-by-point responses to the issues raised by the reviewers:

- We have repeated the experiments shown in **Fig. 1c, 1d, 1f, and Fig. 2b** to obtain a sufficient sample size. ---related to Reviewer #3's comments 1, 2, and 4.
- We have repeated the immunofluorescence staining and added negative controls. The images were replaced by new ones in **Fig. 2a and Extended Data Fig. 2a**. ---related to Reviewer #2's comments 1.
- To further address the features of the high non-CG methylation regions, we have examined the transcriptional changes and preferred dinucleotide sequence (CA, CC, or CT) in the double mutant germ cells. (**Fig. 6e, Extended Data Fig. 7d**). ---related to Reviewer #1's comment 2 and Reviewer #3's comment 10.
- We have added the bisulfite conversion error rate of each WGBS data and correlation coefficient between the biological replicates (**Supplementary Table 2**). ---related to Reviewer #3's comment 6 and 9.
- In addition, we have generated new **Extended Data Fig. 5c** and **Extended Data Fig. 7b**, and modified **Fig. 1a, Extended Data Fig. 1a**, and **Extended Data Fig. 1c** for additional evidence or clarity according to the reviewers' recommendations.

Below please find our responses to the reviewers' comments:

Reviewer #1:

The study addresses the role in vivo of the ADD domains of the de novo DNA methyltransferase DNMT3A and its non-catalytic partner DNMT3L. It does so in the context of male and female germlines in mice: in the female germline, both DNMT3A and DNMT3L are essential for global de novo DNA methylation; in the male germline, both proteins have major roles, but there is more redundancy because of DNMT3B and DNMT3C. The ADD domain is thought to be a major regulator of the functional interaction of DNMT3 proteins with chromatin, as it binds unmethylated lysine 4 of histone H3 (H3K4me0), and this interaction is required to alleviate autoinhibition of catalytic activity. Previous work from this group had generated a DNMT3A-ADD mutant thought to abrogate the interaction with H3K4me0 and evaluated its effect during male and female gametogenesis. In the present study, the authors have generated an analogous DNMT3L-ADD mutant and tested its effect during male and female gametogenesis in isolation and in combination with the DNMT3A-ADD mutant.

In brief, the authors find that:

1. the ADD mutations cause reduced association of DNMT3A and DNMT3L on chromatin in fully-grown oocytes (FGOs), as assessed by immunofluorescence.
2. the ADD mutations cause reduced CG-methylation in FGOs, which is most severe in the double mutants, essentially equivalent to lack of de novo methylation. In spermatogonia, the effects on CG methylation are more modest, probably because of the partial redundancy with DNMT3B and DNMT3C.
3. scRNA-seq analysis of FGOs indicates limited effect on gene expression, with a small number of differentially expressed genes (DEGs) identified, most of which are down-regulated.
4. embryos developing from double-mutant oocytes fail at mid-gestation and display abnormal expression of imprinted genes, consistent with the observed impaired de novo methylation of their germline DMRs in oocytes.
5. non-CG methylation, which is known to be high in FGOs, is also severely affected in the double mutants; however, the authors detect sites of enhanced non-CG methylation (also in spermatogonia), which show enrichment in CA repeats. The authors speculate that these repeats may become sites of recruitment of DNMT3A/3L when the normal recruitment to H3K4me0 is abrogated.

In general, this is a very nicely conducted study, the data are strong and analysed robustly, and the manuscript is presented clearly and succinctly. The authors are able to infer that the ADD domains appears dominant for DNMT3A/3L interaction with and function on chromatin: presence of an intact PWWP domain directed towards H3K36me2/me3 domains is insufficient for efficient de novo methylation in oocytes. On the other hand, other molecular findings are not provided with mechanistic insights.

I have a few minor suggestions:

Response:

We would like to thank the reviewer for the positive remarks on our work! According to the reviewer's suggestions, we have revised the manuscript by adding new observations and computational analyses on the double mutant FGOs. We believe that these new data further strengthened the original conclusions.

1. The authors conclude there is no impact of the Dnmt3a/Dnmt3L-double mutant on oocyte development from the normal gross morphology of ovaries and from the ability to isolate comparable numbers of FGOs which are grossly normal in appearance (bright field). A slightly deeper analysis could be considered; for example, checking nuclear maturation rate (proportion of SN versus NSN oocytes). I just have in mind the transcriptome changes in FGOs from all mutant genotypes, and whether characteristics such as nuclear maturation could contribute to transcriptome differences.

Response:

We appreciate the reviewer's constructive suggestion. We have checked the DAPI-stained wild-type and double mutant FGOs and counted SN-type and NSN-type nuclei (SN is the more mature type). The SN ratio (percentage of SN-type in SN-type plus NSN-type) was not much different between the wild-type (49.0%) and double mutant FGOs (46.2%), which suggests that the transcriptional changes detected in the double mutant FGOs (Figure 5) have virtually no association with nuclear maturation. We have added the data in **Extended Data Fig. 1c**.

2. In some places, the analysis could be deeper to aid mechanistic inferences. For example, the authors do not present much analysis of the FGO scRNA-seq data. There are essentially comparable numbers of DEGs called in the three mutant genotypes (276 in the Dnmt3L-ADD mutant, 421 in the Dnmt3A-ADD mutant, 349 in the double mutant), but in all cases there is an excess of down-regulated genes, suggesting a biological effect rather than false discovery, but for which the authors provide no explanation or speculation. I do not anticipate there will be much useful signal from GO analysis. But it would be useful to know whether there is any overlap amongst the DEGs between the three genotypes.

It is a general observation that substantial loss of CG-methylation in mouse oocytes has very limited impact on gene transcription, but aberrant gains of methylation could perhaps result in impaired transcription. Therefore, it would be interesting to explore whether there were any associated between down-regulated genes and sites of enhanced non-CG methylation.

Response:

We are grateful for the helpful suggestions. We have checked the overlapping DEGs in the three mutants and found that certain fractions of them overlap each other with relatively high odds ratios (**new Extended Data Fig. 5c**). We have added sentences about this in **lines 188–191**.

Next, we addressed whether the genes associated with the high non-CG methylation regions (distance from the TSS ≤ 1 kb; 1427 genes) were down-regulated or not. We found that only a small subset of genes (29 genes) was down-regulated in the double mutant FGOs (**new Extended Data Fig. 7d**). We have added a sentence to describe this in **lines 250–251**.

Reviewer #2:

The de novo DNA methyltransferase Dnmt3a and its accessory factor Dnmt3L are both essential in establishing the DNA methylation landscapes in mouse germ cells. In this manuscript, Kubo et al. provided genetic evidence that the ADD domains of Dnmt3a and Dnmt3L have both distinct and redundant functions in this regard. The data are convincing, and the finding is significant in understanding the mechanisms of action by the DNA methylation machinery in germ cells.

I have a few minor suggestions/comments for the authors to consider:

Response:

We thank the reviewer for the positive remarks and constructive suggestions. We believe that the revisions guided by the reviewer's comments improved our manuscript.

1. Fig. 2a and Extended Data Fig. 2a. The Dnmt3a signal is present in smaller areas in the cytoplasm compared to the Dnmt3L signal, except in double homozygous oocytes. Please comment.

Response:

We have repeated the immunofluorescence staining and confirmed that the DNMT3A signal was present in smaller areas in the cytoplasm compared to the DNMT3L signal. This was observed in all types of FGOs including double homozygous FGOs: so, the double mutants were not an exception. We have also prepared negative controls that were treated similarly but with no primary antibodies and found that the areas of signals were unchanged, suggesting that the signals reflect the real distributions of DNMT3A and DNMT3L (**Extended Data Fig. 2a**). To avoid confusion, we have replaced the double mutant panels in **Fig. 2a** and added images of new replicates in **Extended Data Fig. 2a**.

2. Page 6, lines 138-141: For the statement "Notably, when both DNMT3A-ADD and DNMT3L-ADD had mutations, the global CG methylation level was much more reduced ([Dnmt3aADD/ADD, Dnmt3LADD/ADD], 6.2%), and to our surprise, comparable with that of DNMT3A or DNMT3L knockout FGOs (Fig. 3a, b, c, Extended Data Fig. 3a)", Fig. 3c should NOT be cited.

Response:

We thank the reviewer for pointing out this mistake. We have removed "Fig. 3c" from the sentence.

3. Fig. 3a, a small number of CpG sites normally hypomethylated apparently gained methylation in double homozygous FGOs. From subsequent results and description, I'd guess that those sites were

in the regions that gained non-CpG methylation. If so, I'd suggest that, when describing Fig. 3, the gain of methylation at a small number of CpG sites be mentioned, and "(see below)" can be added to avoid the impression that the data were ignored. If those CpG sites are not in regions with gain of non-CpG methylation, where are they located, and are there general sequence contexts or features?

Response:

We appreciate the reviewer's suggestion. As the reviewer points out, a small number of 10-kb bins normally hypomethylated gained methylation in double homozygous FGOs (Fig. 3a). We now describe this observation in **lines 145–146**, with "see below for more details" as suggested by this reviewer. Furthermore, this reviewer is correct in saying that many of the regions that gained CG methylation in the double mutant FGOs overlapped with the high non-CG methylation regions. We now show this data in **new Extended Data Fig. 7b** and discuss it in **lines 235-237**.

4. Page 11-12, lines 273-277: "In addition, the degree of reduction of immunofluorescence signal was almost equivalent to that observed in [Dnmt3aADD/ADD, Dnmt3LADD/ADD] FGOs, although the degrees of CG methylation loss were significantly different as described above. Because immunofluorescence staining is not quite quantitative, a highly sensitive method to quantify chromatin-bound proteins would be required to resolve the discrepancy". The last sentence appears to suggest that the degree of reduction of immunofluorescence signal could be more severe in double homozygous FGOs, but the difference could not be detected by the non-quantitative method. While that could be one possibility, another possibility is that the degree of reduction was indeed equivalent. In that case, I'd like to hear alternative/additional explanations.

Response:

We are sorry that our statement was not precise. What we meant in the first clause of the last sentence was that immunofluorescence staining is not quantitative and not a direct measure for chromatin-binding, of which the second part was missing. Thus, it was not our intention to suggest that the actual reduction was severer in the double mutant FGOs. Then, one obvious answer to the reviewer's question is that, even if the reduction of immunofluorescence signal is similar, actual chromatin binding could be different. Other possible explanations include that the ADD mutations in both proteins could affect their mutual interaction or the catalytic activity of DNMT3A, perhaps through conformational changes, without affecting their localization, but these are speculations. We have now corrected the statements in **lines 293–296**.

Reviewer #3:

Kubo et al. have systematically investigated the functional role of an intact ADD domain in DNMT3A and DNMT3L in mouse development. One strength of this paper is that the authors did not delete the entire domain, with undefined effects on the functional integrity of the remaining protein parts, but they introduced defined point mutations that disrupt the interaction of the ADD domains with H3-tails. The reported data are interesting and partially surprising. On the more negative side, novelty is compromised by the authors own report of the DNMT3A ADD mutant, which covers 25% of the data included here. Moreover, the paper is largely descriptive and not many mechanistic insights are provided and some technical and conceptual questions are left unanswered.

Response:

We would like to thank the reviewer for providing many useful suggestions. While it is true that we previously reported the role of the DNMT3A ADD in oocytes, the current manuscript reports its role in male germ cells, and the DNMT3L ADD's role in both male and female germ cells, and furthermore the combined roles of the ADDs of the two proteins in both male and female germ cells. Also, the current report only uses the published data for comparisons and never reports the previous findings as new. Therefore, the reviewer's statement that "the authors own report covers 25% of the data included here" is simply untrue. Thanks to the reviewers' suggestions, we have now thoroughly revised the manuscript, highlighting the novel findings and future directions more clearly.

Major comments:

1) I am a bit concerned that the number of animals used for Fig. 1D is low. This should be increased.

Response:

We thank the reviewer for the suggestion. We have done our best to increase the number of crosses and obtained 16 double homozygous mice (8 males and 8 females) (originally 5 males and 4 females). As the probability of obtaining a double homozygous male or female is theoretically one in 32 (but less in the actual crosses; **Fig. 1c**), this is what we could do best.

2) Statistics should be shown for Fig. 1f to justify the corresponding statements in the text.

Response:

We have now obtained statistical values by measuring the long axis of the ovary and testis. The graphs and statistical values are **added to Fig. 1f**.

3) Heterochromatic localization: Literature should be cited for heterochromatic localization of DNMT3A/3L. In this paragraph the term heterochromatin/heterochromatic should be used throughout (lines 112, 116, 118, 120, 124, 127).

Response:

We appreciate the reviewer's suggestion. Based on his/her comments, we now use the term heterochromatin/heterochromatic throughout this paragraph (**lines 114, 116–119, 120, 124, 128, and 130**). Regarding the literature to be cited, while we find many papers describing heterochromatic localization of DNMT3A and DNMT3L signals in ES cells and/or somatic cells, none reported it in FGOs, although their actual images show greater intensities in heterochromatin (Uysal et al., 2017, PMID: 29027601; Hirasawa et al., 2008, PMID: 18559477; Ma et al., 2015, PMID: 26586441). Please also see our response to the major comment 5 below.

4) The number of FGO analyzed in Fig. 2 is not sufficient to draw strong conclusions.

Response:

We have repeated the experiment to increase the number of FGOs for the analyses. The number of double homozygous FGOs has now tripled, by sacrificing 3 more precious females, and those of the other genotypes have also increased.

5) Apparently, the results presented in Fig. 2 do not fit to the rest of the data, as heterochromatic localization was lost for all mutants (including the double mutant) equally, but strongest methylation changes were observed in double mutants. In relation to this there is a conceptual mistake in lines 133 and 246, because Fig. 2 only shows reduced heterochromatic localization but not reduced chromatin binding.

Response:

We are grateful for this insightful comment. Regarding the conceptual mistake issue, we apologize that our wording was not precise: we should have used nuclear or heterochromatic localization of the signals instead of chromatin association of the proteins. To avoid confusion, we have revised the descriptions related to this in **lines 114, 116–119, 120, 124, 128, 130, 136–137, 260, 289, 290, and 294–295**. Regarding the apparent discrepancy between the immunofluorescence data and methylation data, even if the reduction of immunofluorescence signal is equivalent, actual chromatin binding could be different (see our response to comment 4 of Reviewer #2). Other possible explanations include that the ADD mutations in both proteins could affect their mutual interaction or

the catalytic activity of DNMT3A, perhaps through conformational changes, without affecting their localization, but these are just speculations.

6) One critical question regarding Fig. 3 and all the methylation related data is about reproducibility. At least the analysis should be repeated with WT and double mutant and the resulting variance documented.

Response:

Our WGBS datasets have biological replicates, and each dataset exhibits high reproducibility between the replicates (**Figure A**). Only exception is the wild-type spermatozoa, which was only one sample but shows high consistency with our published data (Kubo et al., 2015, PMID: 26290333). The correlation coefficients are now added to **Supplementary Table 2**.

Figure A. Reproducibility of biological replicates of the WGBS datasets.

Scatter plots comparing the CG methylation levels of 10-kb genomic bins between biological replicates in each indicated genotype of FGOs and spermatozoa. Correlation coefficient (r) is also shown.

7) Additional experiments should be conducted to determine if other DNMT3 enzymes, most likely DNMT3C, are affecting the data obtained in spermatozoa. At present the data set is inconclusive. Breeding may be performed with heterozygous DNMT3C KO animals.

Response:

The purpose of this study is to determine the roles of the ADD domains of DNMT3A and DNMT3L in mouse germ cells: therefore, while we agree that determining the enzyme(s) involved in the remaining methylation in the double homozygous spermatozoa may add some useful information, we feel that it is beyond the scope of this manuscript. In addition, obtaining Dnmt3c KO mice and crossing them with our mutant mice should take months, and furthermore the probability of obtaining triple mutant [Dnmt3aADD/ADD, Dnmt3LADD/ADD, Dnmt3cKO/KO] males is only 1/64 or less, which means that

the suggested experiment is not feasible within a reasonable timeframe. The best that we can do at the moment is to compare the CG methylation data from Dnmt3c KO male germ cells at P10 (Barau et al., 2016, PMID: 27856912) with that of our double mutant spermatozoa. Because Dnmt3c KO affects only young L1 retrotransposons with virtually no change in other genomic regions, we focused on the L1 sequences. As shown in **Figure B**, a specific set of young L1 (L1Md_A, L1Md_T, and L1Md_Gf) was more severely affected in the Dnmt3c KO male germ cells than in the double mutant spermatozoa (red circle). This suggests that at least some sequences are more dependent on DNMT3C than DNMT3A/DNMT3L in male germ cells. However, there are other possibilities including the role of DNMT3B, as already discussed in the original manuscript, we decided not to include the data in the revised manuscript.

Figure B

Figure B. Changes of CG methylation levels upon Dnmt3C loss and DNMT3A/3L-ADD loss.

Scatter plots showing changes of CG methylation levels upon loss-of-function of ADD domains of DNMT3A/DNMT3L in spermatozoa (x-axis) and changes upon loss of DNMT3C in P10 male germ cells (y-axis) at young L1 retrotransposons (L1Md_A, L1Md_T, and L1Md_Gf). Genomic regions that lost CG methylation only upon DNMT3C loss are circled in red.

8) In Fig. 5B, the question appears, if the authors conducted MCA without including the MII oocytes as an outgroup. Did the authors test MII oocytes from double mutants?

Response:

According to the reviewer’s suggestion, we have performed the same analysis without including the wild-type MII oocyte datasets (**Figure C**). The MCA pattern appears very similar to the original Figure 5b, except that it has become upside-down. We do not have gene expression data of double mutant MII oocytes because we prioritized other experiments with the precious double mutant females.

Figure C

Figure C. Multi-dimensional scaling without the wild-type MII oocyte datasets.

Multi-dimensional scaling plots of the gene expression profiles of FGOs of the indicated genotypes. Each color-coded dot shows single-cell data.

9) The increase in non-CpG methylation in double mutants is one of the most striking and surprising findings of this work. However, many aspects related to this observation remained unclear and I am not convinced by the mechanistic model proposed by the authors to explain this observation. The double mutant WGBS analysis should be repeated (together with internal controls of unmethylated DNA) to exclude incomplete conversion as a reason for the observations. Incomplete conversion would easily explain the enrichment of non-CpG at poly-CA regions, because at these sites, the CpA density is highest, hence the chances to observe a randomly occurring incomplete conversion in CpA context are highest. Alternatively, the poly-CA sequence may be related with conversion problem, perhaps by forming strong secondary structures.

Response:

We appreciate the reviewer's comments on the observed high non-CG methylation. Regarding the possibility of conversion errors, all our datasets had unmethylated lambda DNA as an internal control, as described in the Materials and Methods section, and the error rates were equally low (0.4–0.9%) in all samples (**Supplementary Table 2, lines 240–242**), even lower than the very well-cited public dataset (Kobayashi et al., 2012, PMID: 22242016) (**Figure D**). Also, we can safely rule out the conversion errors due to the local sequences or secondary structures because the same sequences did not show high non-CG methylation levels in other samples (wild-type and single homozygous mutants). Therefore, the observed high non-CG methylation is not an experimental artefact and is a combined effect of the ADD mutations of the two proteins, although we can only speculate the mechanism. We have added some statements related to these points in the main text (**lines 247–249, 254–256, and 321–322**).

Figure D

Figure D. Conversion error rates of our WGBS datasets.

Barplots showing conversion error rates of all WGBS replicate datasets. Previously published wild-type spermatozoa datasets (Kubo et al., 2015 and Kobayashi et al., 2012) are also shown.

10) Given the known specificity of DNMT3A, the non-CpG methylation should be mainly CpA and this should be shown.

Response:

We appreciate the reviewer's suggestion. We have analyzed the proportions of mCA, mCT, and mCC in wild-type and double mutant germ cells and found that mCA is indeed dominant especially in wild-type oocytes. However, the preference for mCA is compromised in the double mutant cells. We now show this data in **new Fig. 6e** and describe it in **lines 247–249**.

Minor comments

11) It would be good to mention the interaction of DNMT3A1 via its UDR with H2AK119ub1 in the introduction as well (PMID: 33986537), and show the UDR in Fig. 1A as well.

Response:

We now mention the UDR- H2AK119ub interaction in **lines 48–51** and cited Weinberg et al. paper (PMID: 33986537). We also revised **Fig. 1a and Extended Data Fig. 1a**, showing the UDR domain in DNMT3A1.

12) Fig. 1C is difficult to read. Perhaps it would help to draw observed/expected ratios, then readers to not need to do the mendelian calculations themselves.

Response:

We thank the reviewer for the suggestion. As **Fig. 1c** is already busy, we have added the expected Mendelian ratios of the wild-type and double homozygous mice in the figure legend.

13) Where are the regions plotted in Fig. 6F in the WT panels in Fig. 6A and B? Same violin plots as in 6F should be shown for the double mutants.

Response:

As the regions exhibiting high non-CG methylation in wild-type FGOs and spermatozoa were small, the plots shown in Fig. 6f (now **Fig. 6g**) were produced using 1-kb bins (as indicated in the figure), instead of 10-kb bins, which are used to generate Figures 6a and 6b. Keeping this in mind, we have arbitrarily marked in red the 10-kb dots containing the high non-CG methylation 1-kb bins in the scatterplots of Figure 6a and 6b (**Figure E**). Violin plots showing the distribution of the red and black dots along the non-CG methylation levels in wild-type cells are also provided at the bottom. While we agree that this artificial figure may give some additional view, we believe that the important messages

are already provided in **Fig. 6g**. Regarding the second request from this reviewer, since the 1-kb bins used to generate Fig. 6g were selected based on the non-CG methylation level in the double mutant cells (> 20% higher than wild-type), violin plots similar to the one in Fig. 6g, showing the arbitrarily defined values in the double mutants, will not give much information.

Figure E

Figure E. Plotting of high mCH regions identified in previous Figure 6f on the scatter plots of Figure 6a and 6b.

The 10-kb bins that contained the high mCH 1-kb bins identified in previous Figure 6f are marked in red. Violin plots showing the distributions of the red and black dots along the non-CG methylation levels in WT cells are also shown at the bottom.

14) There is a conceptual mistake in the discussion (line 264) because the ADD and PWWP domains will target DNMT3A to distinct genomic regions. Hence, PWWP can never “compensate” for the loss of ADD function.

Response:

We apologize that our explanation was incomplete. The PWWP domain interacts with histone H3K36me2/3, which normally coexists with H3K4me0, and this is why we hypothesized that the PWWP domain could partially compensate for the loss of the ADD function. We have added a phrase to make this point clear (**line 276**).

15) It should be considered, at least in discussion, that the ADD domain of DNMT3A mediates interaction also with other proteins (not only H3) (see for example: PMID 16322236, 30102379, 16322236, 15616584) and at least in the case of MECP2 the interaction has been mapped to the same interface as the H3-tail interaction (PMID: 36519786). Hence the mutated ADD domains studied by Kubo et al. have not only lost interaction with H3-tails but also with MECP2.

Response:

We are grateful for the suggestion. We have added a sentence to mention the interaction with other proteins and cited previous studies (PMID30102379, 15616584, 36519786) in the Discussion section (**lines 283–286**). We find that PMID16322236 has been retracted.

Reviewers' Comments:

Reviewer #1:

Remarks to the Author:

I appreciate the authors' considered responses to comments from all three reviewers, and believe the additions made to the manuscript are appropriate, providing some additional clarity where needed.

Reviewer #2:

Remarks to the Author:

My concerns have been satisfactorily addressed.

Reviewer #3:

Remarks to the Author:

In general, I am satisfied with the authors' response to my comments and the changes in the manuscript. This is a very interesting paper reporting important novel data. I have just two comments:

- 1) p. 11: "which normally coexists with H3K4me0" – A reference must be provided for this statement.
- 2) There is no scientific basis for breaking the non-CG in Fig. 6d into CHH and CHG outside of the plant kingdom. It would be better directly showing CG, CA, CT and CC in 6d, then 6e is not even needed.

We are extremely thankful for the reviewers' time and effort in re-reviewing our revised manuscript! We are delighted that they consider that the revised manuscript is improved, and their comments on the previous submission were adequately addressed. Below please find our responses to their additional comments:

Reviewer comments:

Reviewer #1:

I appreciate the authors' considered responses to comments from all three reviewers, and believe the additions made to the manuscript are appropriate, providing some additional clarity where needed.

Reviewer #2:

My concerns have been satisfactorily addressed.

Response: We appreciate the reviewer's positive remarks on the data quality and findings. We are of the view that our in vivo observation of the various impacts of loss-of-function of ADD domains in DNMT3A and/or DNMT3L on DNA methylation landscapes is novel, and these distinct epigenetic alterations and phenotypes observed in female and male germ cells are also interesting.

Reviewer #3:

In general, I am satisfied with the authors' response to my comments and the changes in the manuscript. This is a very interesting paper reporting important novel data. I have just two comments:

1) p. 11: "which normally coexists with H3K4me0" – A reference must be provided for this statement.

2) There is no scientific basis for breaking the non-CG in Fig. 6d into CHH and CHG outside of the plant kingdom. It would be better directly showing CG, CA, CT and CC in 6d, then 6e is not even needed.

Response: Thank you for the positive remarks on the improvements of the revision. We have added references after the statement "which normally coexists with H3K4me0". Regarding the second comment, as suggested by the reviewer, we have removed the classification of CHH and CHG in Fig. 6d. On the other hand, directly displaying CG, CA, CT and CC in Fig. 6d appeared busy, so we have retained Fig. 6e.